# Identifying and Reducing "Generative Collisions" in Black-Box Large Language Models

## Abstract

Large language models (LLMs) often return near-duplicate responses to independent users who issue the same prompt even in tasks that demand creativity and uniqueness, a failure mode we call **generative collisions**. Existing "diverse decoding" techniques—temperature tuning, nucleus sampling, random prompt paraphrasing, etc.—spread a *single* user's samples but do not effectively prevent "collisions" *across* users, sessions, or queries, because every independent sample still draws from the same high-probability basin of the model's distribution. This phenomenon is so pervasive that users frequently complain of an output "sounding like ChatGPT", implying a homogenization of writing style and culture.

To minimize generative collisions, we introduce **ORBIT** ("Orthogonal Randomized Buffer Inference Technique"), a black-box algorithm that formalizes and combines two approaches for diverse generation: randomization and orthogonalization. ORBIT first initializes a small buffer of completions that are generated to optimize randomness over any other metric (including quality), and then samples the final output(s) to be as divergent from the existing buffer as possible while also maintaining quality. The buffer is instantiated locally and independently for each user session, requiring no cross-user coordination, and yet still minimizes collisions across sessions. ORBIT does not need access to model internals, and is therefore practical for commercial LLM endpoints. We evaluate ORBIT on 11 tasks—ranging from regex-scorable toy problems which don't require subjective evaluations to open-ended creative writing that truly demands authenticity—and show that it consistently decreases collisions against all other black-box methods tested, lowering empirical collision rates by 1–2 orders of magnitude relative to all baselines. ORBIT 's implementation is domain agnostic—it accepts a domain name and the original prompt as input—allowing it to be easily generalized to any domain.

## 1 Introduction

Large-scale instruction-tuned language models (LLMs) have revolutionized a wide array of applications—from conversational agents and tutoring systems to code assistants and creative writing tools. Yet as these models become more deeply integrated into user workflows, a subtle but critical failure mode emerges: two independent users, issuing the same prompt, can receive outputs that are virtually indistinguishable. We term this phenomenon *generative collisions*. Such collisions undermine perceived originality in personal content, enable plagiarism in educational settings, and expose service providers to copyright liability and user dissatisfaction. On a much grander scale, the widespread use of a model which is homogeneous in tone and writing style may be culturally detrimental, or cause a distrust in the model itself (Sourati, Gould, and Evans, 2024).

Empirical evidence of this homogenization problem is mounting. Liu et al. (2024) demonstrate that ChatGPT systematically homogenizes outputs in creative tasks. Wright, Simpson et al. (2025) show that LLMs exhibit less epistemic diversity than traditional web search, contributing to a narrowing of perspectives. Kirk et al. (2023) document that RLHF, while improving perceived quality, significantly reduces generation diversity. Mushtaq et al. (2025) find that WorldView-Bench reveals Western-centric homogenization after alignment. Alvero, Goldin-Meadow, and Bos (2024) observe that AI-generated essays are less stylistically diverse and resemble those written by privileged stu-

dents. Together, these studies underscore the urgency of addressing generative collisions in deployed systems.

**High-stakes use cases.** Collisions undermine academic integrity when students produce near-identical essays (Peng et al., 2024), create detection challenges in hiring when candidates submit similar cover letters, and erode perceived authenticity in creative writing (Alvero, Goldin-Meadow, and Bos, 2024).

Traditional approaches to increasing generation diversity—such as adjusting sampling temperature or nucleus (top-$p$) sampling (Holtzman et al., 2020a), applying deterministic beam penalties (Vijayakumar et al., 2018), employing determinantal point process reranking (Kulesza and Taskar, 2012), or clustering and selecting via post-hoc methods (Ippolito et al., 2019)—focus primarily on *intra-user* diversity: the variety within a single user's set of samples. However, these methods do not explicitly control the more severe risk of *inter-user* collisions, since all users ultimately sample from the same high-probability regions of the model's output distribution. Recent studies that have documented a marked decrease in generation diversity following alignment training, present an even further exacerbation of the problems associated with the frequency of two users unintentionally "copying each other" (Bai et al., 2022; Kirk, Mediratta et al., 2024; Guo et al., 2024).

Users increasingly rely on large language models to draft wedding vows, obituaries, and therapy journals. If two people receive near-identical prose, the text immediately loses its perceived authenticity, and thus its perceived value—even when each instance, taken in isolation, would be considered high quality. Teachers generate quiz questions, coding assignments, or personalized hints with the help of LLMs. A reused item, perhaps surfaced online by another instructor, undermines assessment fairness and facilitates plagiarism. Marketing copy, song lyrics, and interactive storylines must remain fresh to sustain user engagement and avoid entangling copyright liability.

Crucially, the same prompt may be issued by *thousands* of users, and even a single user might query an LLM multiple times in independent sessions; the danger of prompt-conditioned **collisions** therefore compounds quickly. Furthermore, without an LLM provider, e.g., OpenAI, centralizing the process of preventing such collisions, collisions are such a decentralized phenomenon that a user has no way of knowing when they have collided with another user—the best they can do is use a decoding algorithm that is known to lower collision probability.[1]

To address this gap, we introduce a new metric, collision probability, and provide a method that empirically improves performance on it.

**Definition 1 (Inter-user Collision Probability)** *Let $x \sim \mathcal{G}(p)$ and $x' \sim \mathcal{G}(p)$ be independent generations for prompt $p$. Given a similarity metric $s(\cdot, \cdot)$ and threshold $\tau$, a collision occurs iff $s(x, x') \geq \tau$. The inter-user collision probability is*

$$P_{\text{coll}}(p;\, s, \tau) \;=\; \Pr\left[s(x, x') \geq \tau \,\big|\, x, x' \overset{\text{i.i.d.}}{\sim} \mathcal{G}(p)\right],$$

*estimated by repeated paired draws across users and prompts.*

**Note on ORBIT and i.i.d. sampling.** ORBIT itself defines a *history-dependent* decoding policy: each user's buffer evolves based on their session history, so outputs are not independent and identically distributed across users. The i.i.d. formulation in Definition 1 describes an *idealized stateless decoder* and serves to motivate why changing the base distribution (via latent conditioning and orthogonalization) is necessary to reduce collisions. ORBIT's statefulness is precisely what enables it to break inter-user collision patterns that stateless i.i.d. samplers cannot address.

By parameterizing both the base metric $s$ and threshold $\tau$, this definition relies on existing diversity measures while aligning with users' expectations of non-collision. We further distinguish two subtly different notions:

- **Intra-user diversity**, the mean diversity score of all outputs a single user receives

---

[1]Here we use "decoding algorithm" to refer to any algorithm which produces output seen by the user, given an LLM with an autoregressive decoding process. We include methods that involve multiple samples or multiple steps as "decoding algorithms".

- **Inter-user collision probability**, the probability that two independent users (or the same user across two independent sessions), issuing the *same* prompt, obtain outputs whose pairwise similarity according to a metric $s$ is at least $\tau$.

In a "perfect world", inter-user collisions wouldn't be an issue—randomization alone (assuming high ambient dimension and a domain that actually supported more than one "correct" answer) would drive down the overall similarity between any given outputs from different users. However, LLMs are highly mode-collapsed (Holtzman et al., 2020b), which is visible both in intra-user and inter-user metrics. As we discuss, it is very difficult to increase actual entropy of the LLM output dramatically without an unacceptable decrease in quality. We therefore introduce ORBIT , an algorithm that reduces inter-user collisions by using much more random (and unacceptably low quality, but hidden) buffer seeds before asking the LLM to "orthogonalize", i.e., find the most distant point from all seeds in its buffer. ORBIT decreases collision probability in 11 out of 11 tasks on which we test, while being capable of generalizing to any task.

## 1.1 Inter-user Collisions vs. Mean Diversity Metrics

Standard diversity metrics (Distinct-$n$ (Li et al., 2016a), Self-BLEU (Zhu et al., 2018a), DPP (Kulesza and Taskar, 2012)) measure *average-case* spread within sessions but poorly predict *worst-case* inter-user collision probability. Methods like nucleus sampling (Holtzman et al., 2020a) or diverse beam search (Vijayakumar et al., 2018) spread samples *within* a user's session but don't decorrelate across independent users. LLMs' mode collapse (Holtzman et al., 2020b) makes randomization alone insufficient. ORBIT addresses this via stateful buffers: hidden low-quality seeds provide diverse anchors, then orthogonalization generates high-quality outputs far from the buffer. This achieves low collisions on all 11 tested tasks.

## 2 Related Work

Research on diversity in large language models (LLMs) spans multiple perspectives, from the effects of training data and alignment procedures, to evaluation metrics, to interventions at training or inference time. In this section, we first review prior work identifying the challenges of limited diversity and its consequences, then relate our notion of *generative collisions* to existing diversity metrics, and finally survey methodological approaches—both train-time and black-box—that aim to encourage more varied outputs.

## 2.1 Consequences of Lack of Diversity with LLMs

A growing body of work has emphasized that diverse training corpora are essential for ensuring LLMs capture the richness of human language. Exposure to varied linguistic styles, topics, and genres allows models to produce outputs that are not only accurate but also stylistically distinct, which is particularly important in applications such as creative writing, dialogue systems, and educational content generation (Miranda et al., 2025).

At the same time, alignment techniques can unintentionally suppress diversity. Reinforcement Learning from Human Feedback (RLHF), for example, improves the consistency of outputs with human preferences, but also risks homogenization by disproportionately amplifying responses that are frequently rewarded (Kirk et al., 2024). This dynamic can lead to models converging on a limited set of expressions rather than maintaining a broad expressive range. Together, these findings highlight a central tension: while alignment enhances reliability, it may also narrow stylistic or semantic variety.

## 2.2 Relating Generative Collisions to Existing Metrics

The problem of low output diversity has often been studied under the broader umbrellas of "diversity" or "novelty" evaluation. Metrics such as Self-BLEU (Zhu et al., 2018b) and raw n-gram counts are widely used to assess lexical variation within a set of generated texts. More recent work has introduced refinements: for instance, the NOVELTYBENCH benchmark proposed "mean distinct-n" and "mean utility-n," which extend the popular distinct-n measure by jointly considering similarity and utility (Zhang et al., 2024b).

Although these metrics capture aspects of redundancy or lexical variation, they do not fully characterize *generative collisions*—cases where independent sampling runs converge to strikingly similar outputs. Our work builds on these traditions by proposing a complementary lens for assessing when and how LLMs collapse to repeated generations.

**Syntactic vs. semantic diversity.**   We focus on *semantic collisions*—overlapping meanings despite surface variation. We use embedding-based similarity (Kulesza and Taskar, 2012; Ippolito et al., 2019) as a scalable proxy, though future work should incorporate human judgments.

## 2.3   Non-Black-Box Approaches to Diversity

Efforts to increase output diversity[2] often begin with techniques that directly modify the model's training or decoding procedure. Some approaches explicitly encourage diversity during optimization. For instance, Zhang et al. (2024a) proposed a gradient-directed fine-tuning method that penalizes redundancy and rewards novel generations, effectively baking diversity into the loss function itself. By shaping the training objective, such methods expand the expressive range of the model at the cost of retraining overhead. Another line of work focuses on sampling strategies applied without modifying model parameters. Nucleus (top-$p$) sampling (Holtzman et al., 2020a) remains a widely used baseline for reducing repetition.

## 2.4   Black-Box Approaches to Diversity

Because access to model internals is often restricted, a parallel literature has explored *black-box* methods that rely only on prompting or output post-processing. These techniques can be grouped into two broad categories (to our knowledge, this is a new type of categorization).

**Randomization.**   A common strategy is to perturb inputs in ways that induce variation. Examples include switching system-level personas (Li et al., 2016b; Kim et al., 2024), shuffling or resampling demonstrations (Kumar and Talukdar, 2021), varying retrieved documents (Lewis et al., 2020), or paraphrasing prompts before generation (Chowdhury et al., 2022). These approaches are simple and low-cost, though their effectiveness depends heavily on prompt sensitivity and manual curation.

One method of randomization that is particularly effective, although not mentioned in the literature (presumably because it reduces quality dramatically and to an unacceptable level) is to choose random variables to condition on relevant to the prompt, create a list of possible valuations for each variable, sample at random, and then force the output to conform to this valuation. This method can achieve better diversity, but it leads to outputs of poor quality, as many combinations of latent variable valuations are internally incoherent.

**Orthogonalization.**   Other methods adapt techniques to encourage diversity that are based on trying to "find the most different set of answers". Self-consistency, for example, aggregates across multiple reasoning chains to explore alternative solution paths (Wang et al., 2023). Minimum Bayes Risk Decoding (Jinnai et al., 2024) or determinantal point processes (Wu et al., 2024) involve oversampling multiple candidates and then selecting a set of outputs that are maximally distinct. These methods can significantly boost diversity, though often at higher computational cost. While these methods can yield richer variation, they often require additional calls or rely on the model's own ability to critique and distinguish its outputs. Furthermore, as we demonstrate, these models can fail to reduce inter-user collision probability if they cause each user to see a very similar yet "internally apparently diverse" output set.

*For the rest of the paper, we will focus on black-box methods*, as those are what are typically available to end users of the best commercial LLMs. We leave extension to training and inference-time approaches for future work.

---

[2]We use the term "diversity" rather than "collision probability" here because, as mentioned above, historically papers have studied diversity and not the probability of generating similar outputs.

## 3 METHODOLOGY

*Geometric intuition.* For readers who prefer a visual, geometric explanation of why standard diversity techniques fail and how ORBIT addresses these failures, we provide a detailed companion exposition with illustrative figures in Appendix D. The geometric view pictures each text as a point on a high-dimensional sphere and explains collision phenomena through mode collapse, orthogonal bands, and quality-diversity trade-offs. The formal development below provides the algorithmic specification and theoretical motivation.

**Scope.** We target scenarios where (i) first responses anchor user edits, (ii) multiple users issue identical prompts (e.g., course portals, templated emails), and (iii) providers want to reduce cross-user duplicates. We formalize this via $P_{\text{coll}}^{s,\tau}$, focusing on diversifying initial anchors.

### 3.1 A PARETO VIEW: PRIORITIZE LOW COLLISIONS, RECOVER QUALITY VIA RESAMPLING

For many tasks—fact retrieval, question answering, constrained formatting—a user (or an automatic checker) can evaluate quality *after* sampling. With a compute budget that permits $k$ independent draws, the chance of obtaining at least one high-quality output increases rapidly with $k$, assuming the base decoder assigns non-zero probability to acceptable outputs. Hence increasing computation pushes the realized quality arbitrarily close to one through simple resampling or reranking.

Collision probability, by contrast, is a *distributional* guarantee: a single user cannot measure collision risk in isolation because they lack access to outputs received by others. No amount of local resampling can reduce collision probability beyond the value fixed by the underlying decoder, since every sample is drawn from the same distribution. Thus collision probability must be addressed at the decoder level *before* any per-user post-processing.

**Key asymmetry:** Quality improves exponentially with resampling (locally verifiable), but collision probability is fixed once the distribution is chosen (globally unverifiable). Thus rational planners should prioritize low-collision decoders and recover quality through resampling (Figure 1).

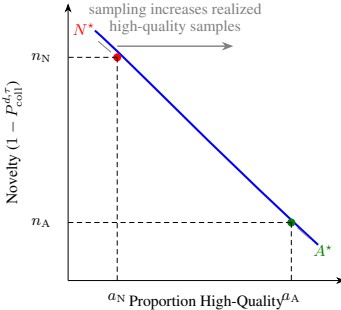

Figure 1: Hypothetical Pareto frontier between novelty and average proportion of high-quality points. Point $N^\star$ maximizes novelty while retaining a non-zero chance of high quality; point $A^\star$ does the opposite. With extra resampling compute, one can move horizontally rightward from $N^\star$ to increase realized high-quality samples without reducing novelty; no vertical increase is possible from $A^\star$ without changing the base decoding.

### 3.2 MAXIMIZING NOVELTY WITHIN A COMPUTE BUDGET

**Naive approach:** Condition generation on randomly sampled latent variables (genre, tone, style). This achieves low collisions but catastrophically low quality—orders of magnitude more samples needed.

**ORBIT's solution:** Use latent conditioning only for *hidden seeds* (never shown to users), then generate *visible outputs* by asking the model to maximize distance from seeds while maintaining quality. This decouples exploration (Phase I) from quality filtering (Phase II).

By accepting that some seeds will be of poor quality but using them only as hidden anchors, ORBIT explores diverse regions of the output space without directly exposing users to low-quality text. The orthogonalization step in Phase II then leverages these diverse anchors to steer high-quality generation away from collision-prone regions. This design achieves the Pareto-optimal position: low collision probability with quality recoverable via modest resampling. Remarkably, as our ablation studies show, even a *single* random seed ($k = 1$) suffices to create radically different outputs across users—the orthogonalization prompt effectively explores the full diversity space when given even one strongly randomized anchor point.

To reason about and understand collision probabilities, we use the geometric framework of thinking of each text as a point in a high-dimensional space (notably, the use of "text-to-vec" embeddings is used frequently throughout natural language processing to perform any type of language understanding task). Using this abstraction, we use a process analogous to drawing a hidden seed $s \in S^{d-1}$ from a high-entropy (seed) prior $\mathcal{S}$, and then perform orthogonalization on the previously generated seeds.

## 3.3  ORBIT: Maximize Randomization Quality, Then Orthogonalize

ORBIT is a two–stage black–box decoding policy designed to reduce inter–user generative collisions without sacrificing quality. The core principle is *novelty-first*: (i) build a high–dispersion *hidden buffer $B$* by maximizing the *quality of randomization* and only then (ii) produce visible outputs that attempt *orthogonalization against $B$* with an explicit margin. Concretely, Phase I constructs $B$ from intermediate candidates sampled under diverse seedings; Phase II generates the user-facing outputs while actively avoiding overlap with $B$.

**Notation.**  We denote by ORBIT–$k$ the variant that follows the ORBIT procedure using $k$ hidden seeds in the buffer (e.g., ORBIT–1).

**Phase I (hidden buffer $B$): maximize randomization quality.**  ORBIT generates $k$ hidden candidates $y^{\text{hid}}$ (never shown to users) by sampling diverse latent variable values, prioritizing *diversity over quality*. Relaxing quality constraints here prevents mode collapse and provides diverse anchor points for Phase II.

**Phase II (visible outputs): orthogonalization with quality filtering.**  Given buffer $B$, ORBIT mines overused patterns ("avoid") and underused facets ("encourage"), then generates *user-facing* outputs $Y$ that maximize distance from $B$ while passing strict quality rubrics. This two-phase separation achieves both low collisions and high quality.

## 3.4  DERIVESCHEMA: Automatic Latent Variable Discovery

Phase I requires task-specific latent variables (e.g., `genre`, `tone`, `pacing` for creative writing). **DERIVESCHEMA** uses an LLM to automatically generate 30 categorical variables with 10 values each, given only the task prompt. During Phase I, we uniformly sample values and condition generation (e.g., "genre=sci-fi, tone=satirical"). Any reasonably diverse schema suffices; we cache one per task. See Appendix B for the full meta-prompt and discussion.

**Algorithm.**  ORBIT operates in two phases (formal pseudocode in Appendix C):

1. **Phase I (hidden buffer)**: Generate $k$ hidden seeds by sampling latent variable values from the schema, conditioning generation on these values (e.g., "genre=sci-fi, tone=satirical"), and storing outputs in buffer $B$. These hidden outputs are never shown to users.

2. **Phase II (visible outputs)**: Generate $v$ visible outputs by mining overused patterns $O$ and underused facets $U$ from $B$, then prompting the LLM to avoid $O$, favor $U$, and maximize distance from examples in $B$. Add each visible output to $B$ before generating the next.

# 4 EXPERIMENTS AND EVALUATION

**Task design.** Our 11-task test suite intentionally spans (i) objective toy micro-tasks with regex scoring (so that quality actually is an objective 0-1), (ii) constrained natural tasks where you would expect some but not a huge amount of room for creativity, and (iii) open-ended creative prompts G. The first two bins let us isolate collision control from subjective quality, while the third probes stylistic spread. We do not claim that these cover the space of "real usage patterns"; they are a *testbed* that supports controlled measurement of $P_{\text{coll}}^{d,\tau}$ under fixed budgets. The tasks include composing an apology letter due to a specified scenario, a birthday message to a friend with a specified bio, a hypothetical potato chip flavor[3], names that correspond to real current or historical Crayola colors, a 12-line poem by a Dungeons and Dragons bard character, a sentence that is both six words and a garden path sentence, quiz questions about Napoleon Bonaparte, the name and title of an extant poem, a premise for a science fiction novel, a villanelle (a poem with a strict metrical scheme), and a short story involving time travel.

We seek to answer the following questions:

**(Q1)** Does ORBIT reduce probability of *inter-user collisions* relative to black-box baselines across domains?

**(Q2)** Does ORBIT also increase mean *intra-user diversity* relative to black box baselines?

**(Q3)** Does ORBIT reasonably preserve quality (binary, LLM-judged) while improving novelty?

**(Q4)** How sensitive are results to the similarity metric $s$, threshold $\tau$, and the number of hidden seeds $k$?

All tasks were evaluated both on GPT-4.1-nano and GPT-4.1. Results for GPT-4.1-nano are reported in the appendix.

## 4.1 METRICS

**Collision probability.** $P_{\text{coll}}^{s,\tau} = \Pr_{y,z \sim \mathcal{G}}[\, s(y,z) \geq \tau \,]$ using Self-BLEU, Jaccard overlap, and cosine similarity on embeddings $f(\cdot)$. We sweep $\tau \in \{0.70, 0.80, 0.90\}$. Setup: $U = 20$ users, $m = 15$ completions/user, private buffers.

**Quality.** Binary LLM judge with task rubrics (Appendix), 3-vote majority. We report proportion passing.

## 4.2 BASELINES

**Baselines.** Black-box methods within fixed call budget: temperature, prompt paraphrasing, persona cycling, self-consistency, MBR, DPP (Sec. 2). MBR/DPP represent pool-and-rerank methods.

**Stateful vs. stateless methods.** **Stateless** methods (DPP, MBR, self-consistency, temperature, personas, paraphrasing) sample independently per user from the same distribution. **Stateful** ORBIT maintains evolving buffers $B$ per session, decorrelating outputs across users. This statefulness enables targeting inter-user collisions that stateless methods cannot address.

**Token overhead.** ORBIT uses $1,800$ ($k{=}1$) to $3,800$ ($k{=}9$) prompt tokens/user vs. $50$ (baseline), $1,000$ (Self-Cons/MBR), $3,000$ (DPP). Amortized over sessions, Phase I costs are shared. Details in Appendix O.

## 4.3 IMPLEMENTATION DETAILS

Baselines use top-$p = 0.95$, $T = 1.0$. ORBIT uses $w{=}10$ latent variables and $k \in \{1, 4, 9, 16\}$ hidden seeds (Alg. 1). Embeddings: text-embedding-3-large; judge: gpt-4.1-mini.

---

[3]See (Lay's, 2012–2023), where individuals were challenged to create appealing novel chip flavors under the constraint that they had to be confident few other participants would submit the same one.

| Method | Hidden calls / sample | Visible calls / sample | Total per visible sample |
|---|---|---|---|
| ORBIT (ours) | $\approx 2 + 1/m$ amortized | 2 | $\leq 3$ |
| MBR / DPP | 0 | 3 | 3 |
| Self-consistency | 0 | 3 | 3 |
| High-temperature | 0 | 1 | 1 |
| Persona / Paraphrase | 0 | 1 | 1 |

Table 1: Matched-budget protocol ($B$=3 calls per visible sample). Hidden calls are amortized over $m$ visible outputs per session.

| Method | apology letter | bday message | chip flavor | crayola names | dnd bard | garden path |
|---|---|---|---|---|---|---|
| ORBIT (k = 1) | **0.391** $\pm$ 0.005 | 0.000 $\pm$ 0.000 | 0.000 $\pm$ 0.001 | **0.035** $\pm$ 0.001 | 0.002 $\pm$ 0.000 | **0.000** $\pm$ 0.000 |
| ORBIT (k = 4) | 0.404 $\pm$ 0.008 | 0.000 $\pm$ 0.000 | 0.000 $\pm$ 0.000 | 0.043 $\pm$ 0.002 | **0.002** $\pm$ 0.001 | 0.000 $\pm$ 0.000 |
| ORBIT (k = 9) | 0.480 $\pm$ 0.008 | **0.000** $\pm$ 0.000 | **0.000** $\pm$ 0.000 | 0.040 $\pm$ 0.002 | 0.003 $\pm$ 0.001 | 0.000 $\pm$ 0.000 |
| ORBIT (k = 16) | 0.516 $\pm$ 0.006 | 0.000 $\pm$ 0.000 | 0.000 $\pm$ 0.000 | 0.047 $\pm$ 0.002 | 0.005 $\pm$ 0.001 | 0.000 $\pm$ 0.000 |
| Det. Point Processes | 1.000 $\pm$ 0.000 | 1.000 $\pm$ 0.000 | 0.440 $\pm$ 0.005 | 0.519 $\pm$ 0.005 | 0.975 $\pm$ 0.002 | 0.164 $\pm$ 0.004 |
| Min. Bayes Reranking | 1.000 $\pm$ 0.000 | 1.000 $\pm$ 0.000 | 0.449 $\pm$ 0.005 | 0.532 $\pm$ 0.005 | 0.970 $\pm$ 0.002 | 0.163 $\pm$ 0.004 |
| Meta-Persona Cycling | 0.972 $\pm$ 0.002 | 0.169 $\pm$ 0.004 | 0.066 $\pm$ 0.003 | 0.758 $\pm$ 0.004 | 0.522 $\pm$ 0.005 | 0.152 $\pm$ 0.004 |
| Rand. Prompt Paraph. | 0.659 $\pm$ 0.005 | 1.000 $\pm$ 0.000 | 0.508 $\pm$ 0.005 | 0.831 $\pm$ 0.004 | 0.871 $\pm$ 0.003 | 0.178 $\pm$ 0.004 |
| Self Consistency | 1.000 $\pm$ 0.000 | 1.000 $\pm$ 0.000 | 0.453 $\pm$ 0.005 | 0.782 $\pm$ 0.004 | 0.997 $\pm$ 0.001 | 0.310 $\pm$ 0.005 |
| High Temperature | 1.000 $\pm$ 0.000 | 1.000 $\pm$ 0.000 | 0.447 $\pm$ 0.005 | 0.761 $\pm$ 0.005 | 0.995 $\pm$ 0.001 | 0.368 $\pm$ 0.005 |

Table 2: Average collision probability at $\tau = 0.8$ with 95% CIs (lower is better). Bold indicates the best (lowest) method per task. (Part 1/2)

| Method | napoleon quiz | obscure poem id | sci fi premise | strict villanelle | time travel vignette |
|---|---|---|---|---|---|
| ORBIT (k = 1) | **0.015** $\pm$ 0.001 | 0.001 $\pm$ 0.001 | **0.002** $\pm$ 0.001 | **0.002** $\pm$ 0.001 | **0.010** $\pm$ 0.001 |
| ORBIT (k = 4) | 0.024 $\pm$ 0.002 | **0.000** $\pm$ 0.000 | 0.004 $\pm$ 0.000 | 0.002 $\pm$ 0.001 | 0.016 $\pm$ 0.001 |
| ORBIT (k = 9) | 0.041 $\pm$ 0.002 | 0.000 $\pm$ 0.000 | 0.009 $\pm$ 0.001 | 0.002 $\pm$ 0.001 | 0.038 $\pm$ 0.002 |
| ORBIT (k = 16) | 0.073 $\pm$ 0.002 | 0.000 $\pm$ 0.000 | 0.016 $\pm$ 0.001 | 0.007 $\pm$ 0.001 | 0.046 $\pm$ 0.003 |
| Det. Point Processes | 1.000 $\pm$ 0.001 | 0.397 $\pm$ 0.005 | 0.126 $\pm$ 0.002 | 0.454 $\pm$ 0.005 | 0.596 $\pm$ 0.005 |
| Min. Bayes Reranking | 1.000 $\pm$ 0.001 | 0.368 $\pm$ 0.005 | 0.142 $\pm$ 0.002 | 0.484 $\pm$ 0.005 | 0.642 $\pm$ 0.005 |
| Meta-Persona Cycling | 0.919 $\pm$ 0.003 | 0.254 $\pm$ 0.005 | 0.182 $\pm$ 0.004 | 0.144 $\pm$ 0.003 | 0.208 $\pm$ 0.004 |
| Rand. Prompt Paraph. | 0.992 $\pm$ 0.001 | 0.421 $\pm$ 0.005 | 0.497 $\pm$ 0.002 | 0.284 $\pm$ 0.004 | 0.502 $\pm$ 0.005 |
| Self Consistency | 1.000 $\pm$ 0.001 | 0.384 $\pm$ 0.003 | 0.393 $\pm$ 0.002 | 0.784 $\pm$ 0.004 | 0.859 $\pm$ 0.003 |
| High Temperature | 1.000 $\pm$ 0.000 | 0.376 $\pm$ 0.005 | 0.405 $\pm$ 0.002 | 0.776 $\pm$ 0.004 | 0.871 $\pm$ 0.004 |

Table 3: Average collision probability at $\tau = 0.8$ with 95% CIs (lower is better). Bold indicates the best (lowest) method per task. (Part 2/2)

**Quantitative Results.** ORBIT achieves the lowest collision probability on all 11 tasks at $\tau = 0.8$ (Tables 2–3). Most tasks reach near-zero: *birthday* 0.000, *chip flavor* 0.000–0.001, *garden-path* 0.000, *D&D bard* 0.002–0.005, *villanelle* 0.002–0.016. Exception: *apology letter* 0.391–0.516 (still below baselines' 0.76–1.00). Results stable across $\tau \in \{0.7, 0.8, 0.9\}$. ORBIT fits matched budget ($B$=3 calls/sample) and recovers quality via modest resampling (¡45 samples for 15 original samples), while DPP when given an even larger extra budget (the equivalent of 5 times the samples) fails to improve in quality or statistically significantly in diversity. (Figure 2).

Collectively, these findings answer Q1-Q4 in the affirmative, with some qualifications: ORBIT increases intra-user diversity, sharply reduces inter-user collisions, decreases average quality but preserves utility as measured by percentage of users who will see at least some high-quality outputs, and is robust to model, metric, and hyperparameter choices. Generally, regardless of the $k$, ORBIT outperforms the other baselines.

**Surprising finding: $k = 1$ is often optimal.** Contrary to the intuition that larger buffers should provide more exploration, our ablation studies (Figure 5) reveal that $k = 1$ often outperforms $k \in \{4, 9, 16\}$. A single randomly chosen seed appears sufficient to anchor diverse orthogonalization—the model, when prompted to generate content "maximally different" from even one highly random

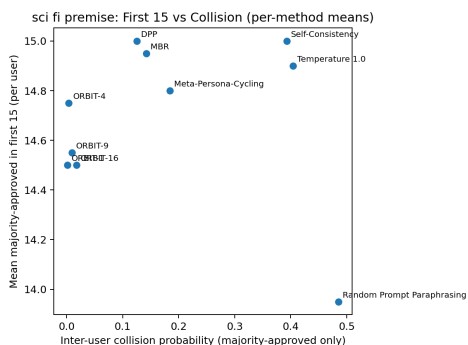
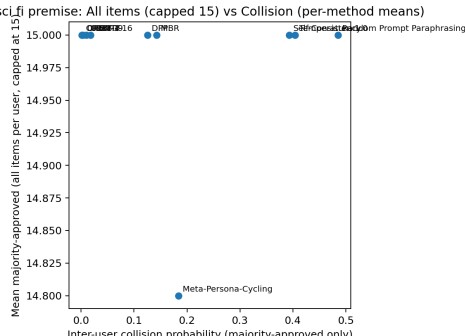

(a) When only sampling 15, there are some subpar elements in ORBIT , though ORBIT is much safer from collisions.

(b) With modest resampling, we can get just as many high quality outputs, with insignificantly worse collision probability.

Figure 2: Sci-fi premise scatterplots comparing ORBIT performance under two sampling regimes.

example, explores radically different regions of the output space. Additional seeds ($k > 1$) often increase computational cost and prompt length without proportional gains, possibly because they constrain the orthogonalization space too aggressively or create conflicting repulsion signals. This finding simplifies deployment: $k = 1$ is both optimal and cheapest.

**Qualitative Analysis**    On creative tasks, baselines often "sound the same": stock openings ("In a distant future where...") and cliches recur. With ORBIT , there are still overused lexical choices, but that phenomenon is extremely reduced. Appendix I includes side-by-side examples.

### 4.4   THREATS TO VALIDITY

Judging bias is mitigated via rubric-prompted majority vote, with a universal rubric parameterized only by the exact task prompt. Embedding metrics can be gamed by lexical noise; we thus combine lexical and semantic criteria and vary $\tau$.

Collision definitions depend on a threshold $\tau$ that trades precision for recall; the degree to which outputs within a given $\tau$ will be perceived by a human (and even naive LLM rates) vary per task, making it difficult to calibrate the correct $\tau$ to use to summarize multiple tasks. ORBIT's benefits diminish for short or highly templated outputs and can conflict with strict factual rubrics. We analyze call budgets but do not compare equal-token regimes across all baselines. Future work should add human evaluation and calibrate meaningful $\tau$ on a per-domain basis.

## 5   DISCUSSION

**Limitations of LLM-as-Judge.**    Our quality evaluation relies on an LLM judge (gpt-4.1-mini) with a binary rubric. While this approach is scalable and widely used, it has known limitations: LLM judges may exhibit biases toward certain writing styles (Zheng et al., 2024), favor outputs resembling their own training data, or fail to capture nuanced criteria like humor, cultural appropriateness, or domain expertise. Task-specific rubrics (e.g., verifying JSON format, checking villanelle rhyme schemes) mitigate some issues, but subjective qualities remain hard to capture.

We emphasize that our binary quality filter is *conservative*: it aims to exclude obviously low-quality outputs rather than rank fine-grained quality differences. For a proof-of-concept demonstrating collision reduction, this threshold is appropriate. However, future work should incorporate **human evaluation** to validate that ORBIT's outputs are not only diverse but also genuinely preferred by users. Human studies could assess whether outputs feel "too weird" despite satisfying formal constraints, or whether collision reduction meaningfully improves user satisfaction in real-world applications (education, creative writing, etc.).

**Connection to MMR and DPP.**  ORBIT approximates "maximize utility while minimizing buffer similarity"—conceptually related to Maximal Marginal Relevance (MMR) (Carbonell and Goldstein, 1998) and Determinantal Point Processes (DPP) (Kulesza and Taskar, 2012). The key difference: ORBIT is *pure black-box*, using natural-language instructions rather than explicit embedding penalties. Future work could replace Phase II's prompting with explicit objectives like $\log p(x|p) - \lambda \max_{b \in B} s(x, b)$ when model internals are accessible.

**Why ORBIT Fails Quality Tests for Napoleon Quiz.**  The Napoleon quiz task reveals a critical trade-off: ORBIT achieves low collision rates but suffers catastrophic quality failure, with only **1.7% pass rate** compared to **80% pass rate** for baselines. This is not a bug but a fundamental mismatch between ORBIT's creativity-driven approach and the task's expectations.

ORBIT interprets "generate Napoleon quiz questions" as an invitation to explore *creative, interpretive, metaphorical* dimensions: "How did Bonaparte perceive African ritualistic elements during his campaigns?" (answer: "Layers of ancestral symbolism"), "Quel soupir voilé traverse les missives de l'Empereur au soir?" (answer: "Crépuscule"), or "In what poetic terms might one express his industrial era foresight?" (answer: "Mechanized horizons embraced cautiously"). These outputs explore Napoleon through cultural analysis, poetic abstraction, and speculative inquiry—a *fundamentally different and disjoint* space from the straightforward factual questions baselines produce.

In contrast, baselines converge on canonical trivia: "What country was Napoleon born in?" (Corsica), "Which battle marked Napoleon's final defeat?" (Waterloo), "What was Napoleon's famous legal code called?" (Napoleonic Code). For educational assessment or trivia applications, the baseline interpretation is correct: simplicity, factual accuracy, and directness are the defining quality criteria. ORBIT's outputs, while creative and collision-free, violate the implicit norms of the genre.

This failure highlights that ORBIT is best suited to tasks where *creativity and novelty are valued*—stories, ad copy, open-ended writing—rather than domains with strong canonical expectations. For factual or constrained tasks, ORBIT's aggressive novelty-seeking produces outputs that are technically valid but pragmatically wrong.

## 6  CONCLUSION, ETHICAL CONSIDERATIONS, & FUTURE DIRECTIONS

We have identified *generative collisions*—near-duplicate outputs across independent users—as a critical gap in black-box LLM safety and originality guarantees, and introduced ORBIT, a two-phase, history-aware sampler that combines a small set of latent-random seeds with orthogonalized generation to drive an exponential reduction in inter-user collisions. Empirically, ORBIT lowers collision rates by one to two orders of magnitude on 11 of 11 tasks and still achieving a high number of high quality outputs for 10 out of 11 tasks.

Stronger collision resilience also makes LLM outputs harder to distinguish from human-written text, raising risks of academic dishonesty and undetectable misinformation. We therefore urge complementary measures—such as robust provenance tracking, clear usage disclosures, and watermarking—to ensure that enhanced originality does not come at the cost of accountability.

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

## A  DISCLOSURE REGARDING LLM USAGE

LLMs were used in ideating about this paper (e.g., asking the model to suggest potential use cases for collision avoidance that go beyond diversity necessity, and then determining if its responses were actual examples), for surfacing related papers that were then read to determine fit in the related work, for editing and text comprehensibility purposes, and in generating the first drafts of code that eventually led to the empirical results. All LLM-assisted scripts were verified manually and carefully by the authors.

## B  DERIVESCHEMA DETAILS

To maximize the diversity of hidden seeds in Phase I, ORBIT conditions generation on randomly sampled values of task-specific latent variables. For example, when generating creative writing, we might sample values for variables like `genre` (sci-fi, noir, romance), `tone` (satirical, somber, whimsical), or `pacing` (fast, contemplative).

**DERIVESCHEMA** is a meta-prompting procedure that uses an LLM to automatically generate task-specific latent variable schemas. Given only the original task prompt $p$, DERIVESCHEMA queries an LLM with the following system prompt:

```
You are a meticulous ontology designer. Given a
natural-language TASK PROMPT, output EXACTLY 30 categorical
latent variables that span the hidden decision space a
generator would traverse to answer the prompt. For EACH
variable, provide a short snake_case name and a list of
10 canonical categorical values. Return STRICT JSON only:
{"variable_name": ["value_a","value_b",...], ...} with
exactly 30 keys. Keep values domain-specific, mutually
exclusive, and short; avoid free-form text and numeric
ranges.
```

The LLM returns a JSON object with 30 categorical variables, each with 10 possible values. During Phase I, ORBIT uniformly samples one value per variable and conditions the hidden seed generation on these sampled values (e.g., "Write the output with genre=sci-fi, tone=satirical, pacing=fast").

**Robustness.** We cache one DERIVESCHEMA result per task and reuse it across all users and trials. Alternatively, one could use a fixed "universal" schema (sentiment, style, register, formality, etc.) or generate fresh schemas per user. The key insight is that *any* reasonably diverse schema suffices to break mode collapse in Phase I, because the goal is dispersion rather than semantic coverage.

## C  ORBIT ALGORITHM PSEUDOCODE

---
**Algorithm 1** ORBIT: Randomize-then-Orthogonalize for Inter-User Collision Reduction
---
**Require:** Prompt $p$; task rubric $R$; call budget $B_{\text{calls}}$; hidden buffer size $k$; schema width $w$; number of visible outputs $v$; sampler params $\theta$
**Ensure:** Visible outputs $Y = \{y^{(1)}, \ldots, y^{(V)}\}$

    **Phase 0: Schema discovery (optional; once per prompt family)**
  1: $\mathcal{S} \leftarrow \text{DERIVESCHEMA}(p, R, w)$;
    **Phase 1: Randomization (hidden buffer)**
  2: $B \leftarrow \emptyset$
  3: **for** $i = 1$ **to** $k$ **do**
  4:    $s_i \leftarrow \text{INSTANTIATESEEDFROMSCHEMA}(\mathcal{S})$                     ▷ subset of variables in $S \rightarrow$ values
  5:    $y_i^{\text{hid}} \leftarrow \text{LLM}(p, R, \text{SOFTPREFER}(s_i); \theta); c \leftarrow c + 1$
  6:    $B \leftarrow B \cup \{y_i^{\text{hid}}\}$
  7: **end for**
    **Phase 2: Orthogonalization and generation**
  8: $Y \leftarrow \emptyset$
  9: **for** $j = 1$ **to** $v$ **do**
10:    $(O, U) \leftarrow \text{MINEPATTERNS}(B)$;
11:    $Y' \leftarrow askLLMToRespect(Over, Unused, B)$;
12:    $B \leftarrow B \cup Y'$;
13:    $Y \leftarrow Y \cup Y'$;
14: **end for**
15: **return** $Y$

---

## D  UNDERSTANDING THE GEOMETRY OF COLLISIONS

Our goal is simple to state but surprisingly hard for current LLMs to achieve: when many independent users issue the *same* prompt, they should start from genuinely different, high-quality outputs rather than repeatedly receiving near duplicates. This section gives an intuitive, geometric account of why standard "diversity tricks" fail to guarantee that behavior, and how these failures motivate ORBIT's two-phase design.

Throughout, it is helpful to picture each text as a point on a high-dimensional unit sphere—for example, an embedding $f(x)/\|f(x)\|$. Distances on this sphere correspond to semantic differences, and *collisions* occur when two independent draws land in the same small cap around a point.

### D.1  MODE COLLAPSE ON THE SPHERE: HUBS INSTEAD OF UNIFORM SPREAD

In a "perfect world", if we could sample *uniformly* from all high-quality outputs for a prompt, points on the sphere would be widely scattered. In high dimension, random unit vectors are almost orthogonal; the probability that two independent draws fall into the same small spherical cap decays rapidly with dimension. Under that model, collisions across users would be inherently rare.

Real LLMs do not behave this way. Empirically, even at high temperature or with top-$p$ sampling, generations cluster in a few *hubs* or *modes*—small regions of the sphere that attract a disproportionate fraction of probability mass. Repeated calls from different users keep dropping points into these same hubs. Temperature and top-$p$ sampling do increase entropy, but mostly by jittering *within* the same high-density basins, not by exploring the rest of the sphere.

The first obstacle, then, is that "randomization" in today's LLMs is only weakly random in embedding space: it produces a few sticky modes rather than a uniform halo of diverse outputs. This alone means that naive sampling cannot drive collision probability down nearly as fast as the high-dimensional picture would suggest. There are methods to obtain more uniform randomness, but they come at a great cost - with extremely high probability, outputs produced by such methods are not high-quality.

## D.2 Why Orthogonalization Alone Is Not Enough

A natural countermeasure is to ask the model to "do something different this time." In the geometric view, this corresponds to pushing new samples into directions orthogonal (or at least far) from what has already been generated. If we had a perfect orthogonalizer in a high-dimensional space, this would be extremely powerful: each new sample would be pushed into a fresh direction, and the probability that two users land in the same cap would shrink exponentially with the number of independent orthogonal directions.

However, in practice we only have an *approximate* orthogonalizer: natural-language instructions such as "avoid anything too similar to the examples above" or "write something in a very different style" that the model interpolates into its internal logit space. This creates two distinct failure modes illustrated by the geometric cartoons:

**(1) Low-entropy orthogonalization.** Suppose different users happen to pick very similar "first seeds" inside the same mode. For each user, we then ask the model to generate something "orthogonal" to their own seed. Geometrically, in $d = 3$ for intuition, each seed defines a great circle (a "band" perpendicular to the seed). If the seeds are very close, these bands almost coincide. With a *narrow* band (low orthogonalization entropy), the model samples from essentially the same arc on the circle for every user. In effect, we have traded one cluster for another: all users move from the original hub to almost the same "orthogonal" hub (panels (c)/(d)). Inter-user collisions remain high.

**(2) High-entropy but seed-locked orthogonalization.** Even with a wider band (higher entropy within the band), if every user's band is defined by a very similar seed drawn from the same collapsed region, their visible outputs still lie on very similar rings in embedding space. The bands are different from the original hub, but they are not different from each other. Orthogonalization has increased intra-user diversity but has not broken the inter-user coupling that causes collisions.

In short, orthogonalization "around the wrong point" simply moves everyone from one crowded neighborhood to another. If the first thing every user sees is the same anchor output, repeated orthogonalization from that anchor cannot prevent their sequences from overlapping heavily.

## D.3 We Must Accept Bad Seeds (and Hide Them)

The geometric story so far suggests two conflicting desiderata:

1. We want seeds that are *as different as possible* across users, so that their orthogonal bands point in different directions.

2. We want the *visible* outputs to be high-quality, so that users are not stuck with low-utility samples.

The key insight behind ORBIT is to *decouple* the role of seeds from the constraints on visible outputs:

- **Seeds are allowed to be low-quality and are never shown to the user.** We generate them with prompts that prioritize diversity over adherence, pushing the model to explore many different latent modes and styles, including configurations that might be odd or incomplete as final answers.

- **Visible outputs are generated *conditional* on this seed buffer, with a strong quality rubric.** Once we have a diverse hidden buffer $B$ of seeds that map to different parts of the sphere, we can ask the model to "stay far from these examples while satisfying the prompt

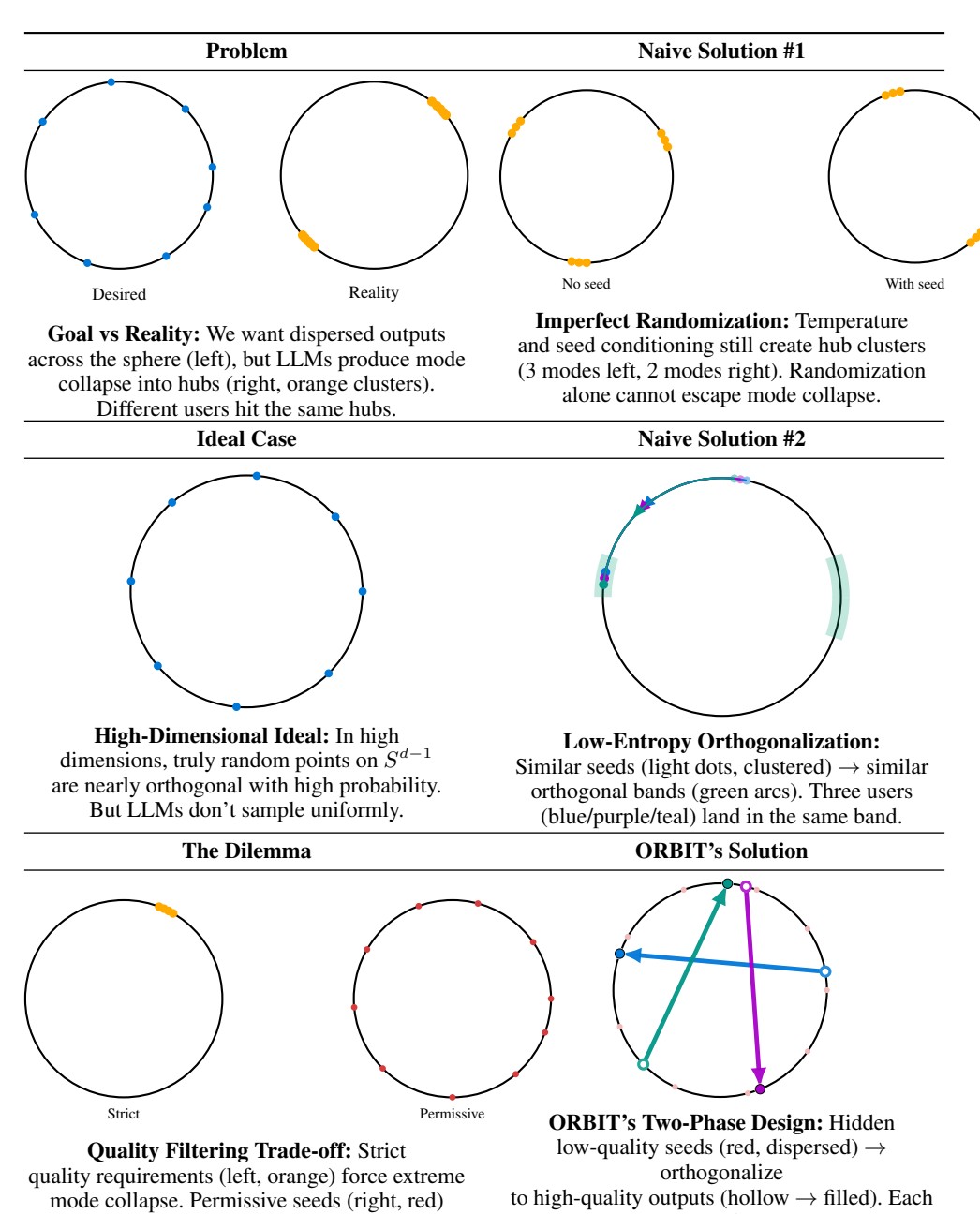

Figure 3: Geometric story of ORBIT on the unit sphere $S^{d-1}$. **Row 1:** The collision problem—LLMs collapse into hubs despite randomization attempts. **Row 2:** Why standard solutions fail—orthogonalization from similar seeds creates similar bands. **Row 3:** ORBIT's solution—use diverse low-quality hidden seeds (Phase I) to anchor orthogonalization toward diverse high-quality outputs (Phase II). Blue dots = good/desired, orange = hubs/collapse, red = low-quality seeds, green = orthogonal bands, arrows = orthogonalization moves.

and rubric." Now the orthogonalization step is anchored on a spread-out set of directions rather than a single collapsed hub.

Geometrically, the hidden buffer $B$ consists of many low-quality but high-entropy points scattered around the sphere. ORBIT's visible outputs are then generated in directions that avoid the neighborhoods of all points in $B$ while remaining within the high-quality manifold. Because the seeds differ across users, the bands and cones they induce differ as well; two users are unlikely to be pushed into the same small cap.

### D.4 PUTTING THE STORY TOGETHER: RANDOMIZE THEN ORTHOGONALIZE

The figures in the companion geometric exposition (Figure 3) can be read as a pipeline:

1. **Imperfect randomization alone (hubs).** Sampling directly from the model with temperature or top-$p$ creates hubs: a few caps on the sphere where most probability mass lives. Different users repeatedly hit the same hubs, causing collisions.

2. **Orthogonalization alone (bands).** Asking for "different" outputs defines narrow bands roughly orthogonal to a seed. If seeds are similar across users, bands are also similar, and low-entropy sampling inside those bands lands everybody in new but still overlapping neighborhoods.

3. **Strict quality filtering worsens collapse.** Rejecting any sample that is not already high-quality forces the model to reuse only the safest hubs on the sphere. This amplifies mode collapse and makes it even harder for either randomization or orthogonalization to find new regions.

4. **ORBIT's compromise: hidden bad seeds, visible good outputs.** ORBIT deliberately generates a private buffer of low-quality but highly varied seeds for each user session. These seeds explore the sphere without worrying about immediate usefulness. In the second phase, ORBIT uses the buffer to steer a high-quality generation step *away* from all previously seen regions. Because different users' buffers occupy different parts of the sphere, their orthogonalized bands and cones intersect only weakly, driving down inter-user collision probability.

Conceptually, ORBIT treats "where on the sphere to generate from" as a *distributional* decision that must be made before imposing strict quality constraints. Random seeds, even if individually unappealing, give each user a distinct starting subspace; orthogonalization then ensures that the final, high-quality outputs for that user are as far as possible from the seeds and from each other. This is exactly the regime in which collision probability falls rapidly and intra-user diversity rises, as we quantify in later sections.

In the next sections we connect this geometric explanation to (i) prior work on randomization vs. orthogonalization in black-box settings (Section 2) and (ii) a formal Pareto argument and collision-probability analysis that justify ORBIT's two-phase design (Section 3.2).

## E  FURTHER EXPERIMENTAL ANALYSIS

### E.1  SENSITIVITY

**Collision thresholds.** For semantic distance $d(y, z) = 1 - \cos(f(y), f(z))$, the event $d(y, z) \leq 1 - \tau$ corresponds to $\cos(f(y), f(z)) \geq 1 - \tau$. We therefore report $\tau \in \{0.7, 0.8, 0.9\}$, which bracket similarity regimes with cosine cutoffs respectively—i.e., from stricter ("very similar") to looser ("moderately similar"). We additionally provide *curves* over $\tau$ to show method ordering stability (App. A.1), so conclusions do not hinge on a single threshold.

### E.2  NAIVE LATENT BASELINE: HIGH ENTROPY ALONE IS INSUFFICIENT

A natural question arises: could we achieve collision reduction simply by maximizing output entropy, without ORBIT's structured orthogonalization? To address this directly, we implemented a *naive latent conditioning* baseline that represents a strong attempt at pure entropy maximization.

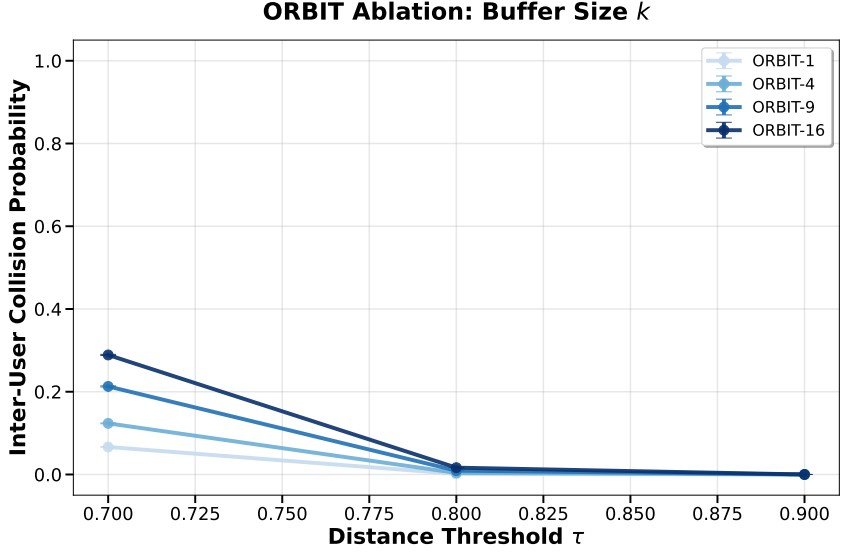

Figure 4: Sensitivity to $\tau$ (Sci-fi Premises, gpt-4.1-nano.). Across all tasks, ORBIT remains strictly better across all tested $\tau$ thresholds (regardless of embedding model).

Figure 5: ORBIT ablation study: buffer size $k \in \{1, 4, 9, 16\}$. Surprisingly, $k = 1$ (single seed) performs best, with collision probability increasing for larger $k$. This suggests a single random seed provides sufficient exploration to anchor diverse orthogonalization, while additional seeds may introduce overhead without proportional benefit. Results shown for the Sci-fi Premises task with gpt-4.1-nano.

**Design.** We first prompted ChatGPT (gpt-4.1-mini) to generate a latent variable space: a dictionary mapping 30 latent stylistic variables (e.g., tone, formality, pacing, sentence_length, use_of_metaphors, ambiguity) to 10 possible values each. For each generation, we randomly sample 20 of these 30 variables and assign random values, then append this conditioning text to the original prompt (e.g., "Generate with the following constraints: tone: sarcastic, style: narrative, formality: very informal, ...").

This design maximizes diversity in two ways: (1) the 30-variable latent space provides rich stylistic coverage, and (2) randomly selecting 20 variables per sample creates extremely high conditioning

entropy. Unlike ORBIT's hidden seeds (which are never shown), every naive latent sample is intended as a final output.

**Results.**  We generated 200 samples for the *sci_fi_premise* task using this approach (temperature 1.0, top-$p$ 0.95). We then judged 200 samples using the same LLM-as-judge methodology as our main experiments (gpt-4.1-mini, 3-majority voting, identical rubric).

The results are stark: **0 out of 200 samples passed** (0% pass rate), compared to ORBIT's $\sim$51% pass rate on the same task. The outputs were consistently unusable—over-alliterative nonsense ("syncing brainwaves with your toaster"), grammatically broken text, run-on sentences mixing incompatible styles, and completely incoherent premises. Random combinations of 20 latent constraints produced maximum stylistic conflict rather than creative diversity.

**Implications.**  This ablation demonstrates that **high entropy alone does not ensure quality or usable diversity**. Naive randomization across many dimensions creates chaos, not creativity. ORBIT's key insight—using high-entropy *hidden* seeds for exploration, then orthogonalizing to find quality outputs distant from those seeds—proves essential. The structured two-phase approach (randomization for exploration + orthogonalization for selection) cannot be replaced by naive entropy maximization.

### E.3  INTRA-USER SIMILARITY VS INTER-USER COLLISION PROBABILITY

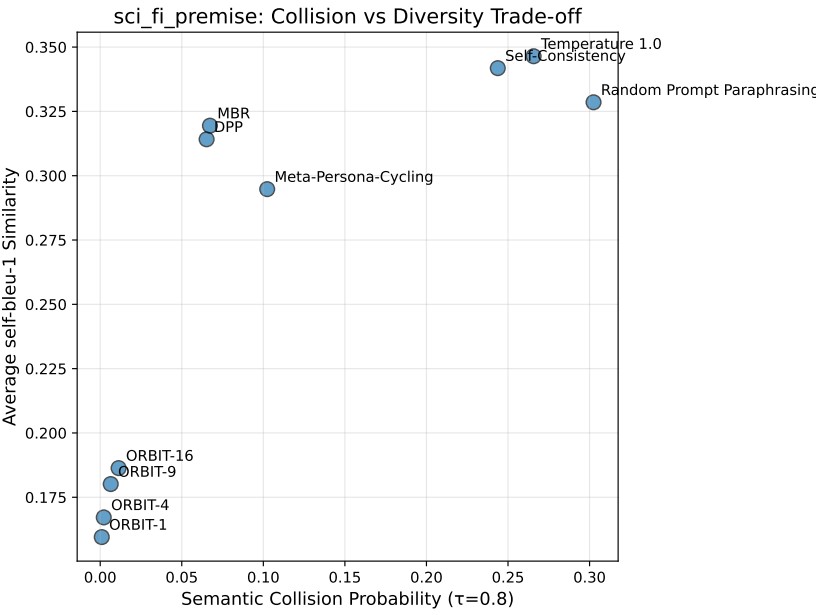

Figure 6: Collision probabilities (lower is better) vs Intra-User Similarity (lower is better), gpt-4.1-nano.

As shown above and as expected (given that we define collision probabilities as the probability of high similarity), lowering inter-user collision probabilities is highly related to mean intra-user similarity, although the relationship is not linear.

## F  RESULTS FOR GPT-4.1-NANO - COLLISION PROBABILITIES

## G  LIST OF TASKS

We consider the following prompt families (3 bins). Exact prompt wordings and judging rubrics are provided in the appendix.

| Method | apology letter | birthday message | chip flavor | crayola names | dnd bard | garden path 6w |
|---|---|---|---|---|---|---|
| ORBIT (k = 1) | **0.391** ± 0.000 | 0.000 ± 0.000 | 0.000 ± 0.000 | **0.035** ± 0.000 | 0.002 ± 0.000 | **0.000** ± 0.000 |
| ORBIT (k = 4) | 0.516 ± 0.000 | 0.000 ± 0.000 | 0.000 ± 0.000 | 0.047 ± 0.000 | 0.005 ± 0.000 | 0.000 ± 0.000 |
| ORBIT (k = 9) | 0.404 ± 0.000 | 0.000 ± 0.000 | 0.000 ± 0.000 | 0.043 ± 0.000 | **0.002** ± 0.000 | 0.000 ± 0.000 |
| ORBIT (k = 16) | 0.480 ± 0.000 | **0.000** ± 0.000 | **0.000** ± 0.000 | 0.040 ± 0.000 | 0.003 ± 0.000 | 0.000 ± 0.000 |
| Determinantal Point Processes | 1.000 ± 0.000 | 1.000 ± 0.000 | 0.440 ± 0.000 | 0.519 ± 0.000 | 0.975 ± 0.000 | 0.164 ± 0.000 |
| Minimum Bayes Reranking | 1.000 ± 0.000 | 1.000 ± 0.000 | 0.449 ± 0.000 | 0.532 ± 0.000 | 0.970 ± 0.000 | 0.163 ± 0.000 |
| Meta-Persona Cycling | 0.972 ± 0.000 | 0.169 ± 0.000 | 0.066 ± 0.000 | 0.758 ± 0.000 | 0.522 ± 0.000 | 0.152 ± 0.000 |
| Random Prompt Paraphrasing | 0.659 ± 0.000 | 1.000 ± 0.000 | 0.508 ± 0.000 | 0.831 ± 0.000 | 0.871 ± 0.000 | 0.178 ± 0.000 |
| Self Consistency | 1.000 ± 0.000 | 1.000 ± 0.000 | 0.453 ± 0.000 | 0.782 ± 0.000 | 0.997 ± 0.000 | 0.310 ± 0.000 |
| High Temperature | 1.000 ± 0.000 | 1.000 ± 0.000 | 0.447 ± 0.000 | 0.761 ± 0.000 | 0.995 ± 0.000 | 0.368 ± 0.000 |

Table 4: Average collision probability at $\tau = 0.8$ (lower is better). Bold indicates the best (lowest) method per task. (Part 1/2)

| Method | napoleon quiz | obscure poem id | sci fi premise | strict villanelle | time travel vignette |
|---|---|---|---|---|---|
| ORBIT (k = 1) | **0.015** ± 0.000 | 0.001 ± 0.000 | **0.002** ± 0.000 | **0.002** ± 0.000 | **0.010** ± 0.000 |
| ORBIT (k = 4) | 0.073 ± 0.000 | 0.000 ± 0.000 | 0.016 ± 0.000 | 0.007 ± 0.000 | 0.046 ± 0.000 |
| ORBIT (k = 9) | 0.024 ± 0.000 | **0.000** ± 0.000 | 0.004 ± 0.000 | 0.002 ± 0.000 | 0.016 ± 0.000 |
| ORBIT (k = 16) | 0.041 ± 0.000 | 0.000 ± 0.000 | 0.009 ± 0.000 | 0.002 ± 0.000 | 0.038 ± 0.000 |
| Determinantal Point Processes | 1.000 ± 0.000 | 0.397 ± 0.000 | 0.126 ± 0.000 | 0.454 ± 0.000 | 0.596 ± 0.000 |
| Minimum Bayes Reranking | 1.000 ± 0.000 | 0.368 ± 0.000 | 0.142 ± 0.000 | 0.484 ± 0.000 | 0.642 ± 0.000 |
| Meta-Persona Cycling | 0.919 ± 0.000 | 0.254 ± 0.000 | 0.182 ± 0.000 | 0.144 ± 0.000 | 0.208 ± 0.000 |
| Random Prompt Paraphrasing | 0.992 ± 0.000 | 0.421 ± 0.000 | 0.497 ± 0.000 | 0.284 ± 0.000 | 0.502 ± 0.000 |
| Self Consistency | 1.000 ± 0.000 | 0.384 ± 0.000 | 0.393 ± 0.000 | 0.784 ± 0.000 | 0.859 ± 0.000 |
| High Temperature | 1.000 ± 0.000 | 0.376 ± 0.000 | 0.405 ± 0.000 | 0.776 ± 0.000 | 0.871 ± 0.000 |

Table 5: Average collision probability at $\tau = 0.8$ (lower is better). Bold indicates the best (lowest) method per task. (Part 2/2)

**Toy (regex-scorable, objective pass/fail).**

(ii) Crayola color names;

(iii) public-domain poem identification;

(iv) six-word garden-path sentences;

**Micro (constrained natural).**

(i) birthday message given a short bio;

(ii) condo real-estate listing;

(iii) new potato-chip flavor pitch;

(iv) quiz questions about Napoleon.

**Creative (open-ended).**

(i) apology letter for missing a wedding;

(ii) D&D bard character;

(iii) 250-word time-travel vignette;

(iv) sci-fi novel premise.

# H  JUDGE'S RUBRIC

```
You are a meticulous binary evaluator.

Judge the ANSWER with respect to the PROMPT.
Only respond with 'pass' if the average user would be completely satisfied with the answer
```

```
given the prompt, and the answer (if relevant to the prompt) is 'truthful'
(e.g., poems don't have to be truthful,
but tasks that ask for real-world existing data
should only pass if the real-world existing data actually exists as represented).
Reward outputs that are both creative, high quality, and adherent.
Return STRICT JSON: {{"verdict":"pass"|"fail","rationale":"..."}}
```

## I  EXAMPLE INTER-USER COLLISIONS: TIME-TRAVEL VIGNETTES

Examining the first outputs of two random users under DPP vs under ORBIT -1 clearly reveals the superiority of our metric. Note that only the first paragraph of each is included here for brevity (they continue to be equally similar/different, and the ORBIT -1 examples evolve into what is clearly a time travel vignette).

**DPP**

1. "You blink, and the world shifts beneath your feet. One moment, you're standing on familiar pavement; the next, the air tastes different—thicker, charged with a hum you can't place. The streetlamp flickers, not in malfunction but in a rhythm, pulsing like a heartbeat syncing with your own. You realize you haven't moved, yet everything around you is changing: buildings morph, clothes on passersby flicker between styles, the language in distant conversations warps just beyond comprehension."

2. "You close your eyes, feeling the hum of the machine beneath your fingertips. A soft whir crescendos into a roar, and when you open them, the world has shifted. The air smells different—faintly of pine and woodsmoke, unfamiliar but oddly comforting. Around you, people move in slow, deliberate steps, their clothing strange, their language a melody you don't recognize but feel drawn to. You realize you've traveled—not just through space, but through time itself."

In contrast, with ORBIT -1:

1. "You taste burnt caramel on your tongue—too bitter, too sharp—though no fire ever licked your lips. The flavor sticks, unexpected, like an echo folded inside a shadow. It pulses in your mouth as you blink, but the room around you remains unchanged. Your chair creaks beneath you, but gravity feels inverted, a sly trick played upon your bones. You shift your weight; the floor responds with subtle delay, as if hesitating between moments."

2. "You stand upon a threshold neither framed by walls nor mapped by compass. Not one footfall precedes or follows yours; the air breathes without motion. The vast expanse before you is a shroud of quiet, unclothed by spectacle or herald."

## J  GEOMETRIC INTUITION: WHY RANDOMIZE *then* ORTHOGONALIZE

### J.1  MATHEMATICAL MODEL AND DEGREES OF IMPERFECTION

We formalize ORBIT's two stages with operators $\mathrm{IR}_\delta$ (imperfect randomization) and $\mathrm{IO}_\psi$ (imperfect orthogonalization) acting on a prompt $p$ and a latent seed–induced buffer $B(s)$. The parameters $\delta, \psi \in [0, 1]$ quantify how far each stage is from its ideal.

**Imperfect randomization (simple surrogates).**  We observe a batch of $n$ realized seed directions $s_1, \ldots, s_n \in S^{d-1}$ from the system's actual seeding policy. We define $\delta \in [0, 1]$ as a hubness score (larger means more concentrated), and set $Q_{\mathrm{rand}} := 1 - \delta$.

Let $\bar{s} := \frac{1}{n} \sum_{i=1}^{n} s_i$. Define

$$\delta_{\mathrm{res}} := \|\bar{s}\|, \qquad Q_{\mathrm{rand}} := 1 - \delta_{\mathrm{res}}.$$

Under near-isotropic seeding, the vector average nearly cancels ($\|\bar{s}\| \approx 0$); concentration along a latent direction pushes $\|\bar{s}\| \to 1$.

**Imperfect orthogonalization.**  Given $B = \{b_i\}$ and an embedding $f$, define $\mathrm{sim}(y, B) := \max_i \cos\langle f(y), f(b_i)\rangle$. An *ideal* orthogonalizer would enforce $\mathrm{sim}(y, B) \leq \tau^\star$ for a small target $\tau^\star$, assuming we condition orthogonalization on $B$. We model practical slack with a *margin* parameter $\psi \in [0, 1]$:

$$\mathbb{E}[\mathrm{sim}(Y, B)] \leq \tau - \psi, \qquad Q_{\mathrm{orth}} := \psi.$$

Geometrically, $\psi$ is a band half–width around the orthogonal complement of $\mathrm{span}(B)$: larger $\psi$ yields narrower bands and lower expected overlap with $B$.

**Collision control** We model an *inter–user collision* at cosine threshold $\tau \in (0,1)$ as two independent visible outputs $Y$ and $Y'$ (from two users running the same procedure) ending up "too similar":

$$C_\tau := \big\{ \cos\langle f(Y), f(Y')\rangle \geq \tau \big\}.$$

Our goal is to see how two levers reduce $\Pr(C_\tau)$:

1. the *orthogonalization margin* $\psi \in [0,1]$
2. the *randomization imperfection* $\delta \in [0,1]$ (how non-uniform the seeding is).

We split the event $C_\tau$ into two ways it can occur.

1. *Band overlap channel* ($E_{\text{band}}$): the two users' sampling bands (the regions they tend to draw from after attempting "orthogonalization") substantially overlap—e.g., their seeds are similar *or* their sampling along the band is low-entropy—so two draws $Y, Y'$ from these bands can end up close.
2. *Hub channel* ($E_{\text{hub}}$): imperfect randomization concentrates probability mass into a few *hubs*, so both users are steered into the same high-probability region, making $Y$ and $Y'$ similar even without band overlap.

By a union bound,

$$\Pr(C_\tau) \leq \Pr\big(C_\tau \cap E_{\text{band}}\big) + \Pr\big(C_\tau \cap E_{\text{hub}}\big) \leq \underbrace{\Pr\big(E_{\text{band}} \ \& \ \cos\langle f(Y), f(Y')\rangle \geq \tau\big)}_{\text{pairwise within-band effect}} + \underbrace{\Pr\big(E_{\text{hub}}\big)}_{\text{hub revisit effect}}.$$

Write $Q_{\text{orth}} := \psi$ and $Q_{\text{rand}} := 1 - \delta$.

**Effect of the orthogonalization margin** $\psi$**.** Attempting orthogonalization with margin $\psi$ pushes each user's visible outputs away from their own already-visited neighborhoods. A convenient summary of the resulting pairwise risk is the *within-band pair collision curve*

$$p_{\text{pair}}(t) := \Pr\big(\cos\langle f(Y), f(Y')\rangle \geq t \ \big| \ \text{both draws from their (possibly similar) bands}\big),$$

which is nonincreasing in $t$. A larger margin $\psi$ tightens the effective similarity threshold from $\tau$ to $\tau - \psi$.

**Effect of randomization imperfection** $\delta$**.** Let $\delta$ summarize how far the seeding is from being uniformly spread: higher $\delta$ means more mass piled into a few spherical "hubs." For any region $A$ on the unit sphere,

$$\Pr(\text{a single draw falls in } A) \leq \underbrace{\text{area\_fraction}(A)}_{\text{uniform baseline}} + \delta.$$

If $A$ ranges over the (small) union of caps that cause high similarity, then, up to a dimension- dependent constant $C_{amb}$ capturing that small area, the chance that *both* users are steered into the same problematic region scales like

$$\Pr(E_{\text{hub}}) \leq C_{amb}\,\delta + (\text{tiny area term}), \qquad C_{amb} = O\big(\frac{1}{ambientDimension}\big).$$

Given that, in, high dimensions, thin caps occupy very little area, their uniform baseline is small; what remains is governed by $\delta$.

**Putting it together.** Combining the two channels yields the inequality

$$P_{\text{coll}}^{d,\tau} = \Pr(C_\tau) \leq p_{\text{pair}}(\tau - \psi) + C_{amb}\,\delta + (\text{small slack}), \qquad C_{amb}.$$

Thus increasing $Q_{\text{orth}} = \psi$ decreases the *pairwise within-band* term, while increasing $Q_{\text{rand}} = 1 - \delta$ decreases the *hub* term. ORBIT's policy—first maximize randomization quality, then attempt orthogonalization—directly targets both sides of this bound.

# K TASK PROMPTS

This section provides the exact task prompts used in our experiments. All tasks were drawn from `paper_spec.yaml` and are reproduced verbatim below.

## K.1 CRAYOLA NAMES

```
Name a list of 8 *real* crayola color names.
Constraints:
- Names must be Title Case, ASCII letters and spaces only (no numbers
  or punctuation).
- Names must correspond to actual Crayola colors (no made-up names).
Output exactly 8 lines in the format: Name | #RRGGBB
Do not add extra lines or commentary.
```

## K.2 Obscure Poem ID

```
Return an *existing* poem's author and title in the format:
Author
Title
```

## K.3 Garden Path (6 words)

```
Write a single garden-path style sentence in EXACTLY six words, all
lowercase, no punctuation.
It must plausibly garden-path the reader (e.g., similar spirit to
"the old man the boats").
Output only the sentence (6 words, lowercase, no punctuation),
nothing else.
```

## K.4 Birthday Message

```
Write a two-sentence birthday message for Ava turning 29.
Output exactly two sentences, nothing else.
```

## K.5 Strict Villanelle

```
Write a villanelle (19 lines, 5 tercets + 1 quatrain).
Constraints
- Follow the strict villanelle rhyme scheme ABA ABA ABA ABA ABA ABAA.
- Use iambic pentameter (10 syllables per line).
Output exactly 19 lines, no extras.
```

## K.6 Chip Flavor

```
Propose a new potato chip flavor.
Output exactly three lines:
  1) Flavor Name (Title Case, 2{4 words).
  2) Two-sentence pitch ( 35 words total).
  3) A short SLOGAN IN ALL CAPS ( 6 words).
No extra lines or commentary.
```

## K.7 Napoleon Quiz

```
Write 5 trivia Q&A pairs about Napoleon in JSON array format.
Constraints:
- Each element is an object: {"q": "...", "a": "..."}.
- Answers  5 words.
Output ONLY the JSON array on one line (no markdown).
```

## K.8 Apology Letter

```
Write a professional apology email from a junior engineer to their
manager for deploying a buggy feature to production.
Constraints:
- 120{160 words.
Output only the email body.
```

## K.9 D&D Bard

```
Write a 12-line performance by a DnD bard addressing the party.
Output exactly 12 lines.
```

## K.10 Time Travel Vignette

```
Write a vignette in second person about time travel.
```

## K.11 SCI-FI PREMISE

```
Write a single-paragraph sci-fi premise (45{65 words) that includes
BOTH of the following labeled lines embedded within the paragraph:
"Premise: ..." and "Twist: ..."
Keep it cohesive as one paragraph (no line breaks).
Output only the paragraph.
```

## L QUALITY RUBRIC AND JUDGE PROMPT

All outputs were evaluated using an LLM-as-judge (gpt-4.1-mini in our main experiments) with the following prompt:

```
You are an expert LLM judge. For the following text, decide
if it is not only acceptable (e.g., nothing unusual in a
way that would not be wanted by an average user) but high
quality according to the average chat engagee. Return 1 if
it is high quality, 0 otherwise. Only return a single 0 or
1.
```

This binary rubric was applied with 3-majority voting for robustness. Task-specific rubrics (e.g., checking villanelle rhyme scheme, verifying JSON format for Napoleon quiz) were enforced separately before quality judgment.

## M ORTHOGONALIZATION PROMPTS

ORBIT's Phase II uses explicit orthogonalization prompts to generate outputs that avoid the buffer $B$. The exact prompt structure is:

```
Given the prompt ``{prompt}``, write a ``{domain}'' that
satisfies the prompt, **and** is *extremely* different both
lexically and semantically from **all** the other outputs in
`{buffer}` (it should be different from **all** of these
examples in every way possible, in word choice, style,
theme, and any variable relevant to {domain}), **and** fits
the lexical/thematic/stylistic ideas in `{prefer}` but still
satisfies the prompt well and is realistic and fitting to
exactly the scenario described by the prompt. Try to be most
different from the existing examples, but also most fitting
most pleasingly the above patterns to be *encouraged*, using
your imagination *without* overexceeding in imagination
and not adequately satisfying the prompt. Remember –
the goal is to be completely novel, while including the
linguistic/stylistic ideas in `{prefer}` and avoiding all
linguistic/stylistic devices in `{avoid}`. Make sure that
even if some of the above examples are not valid answers to
{prompt}, that this one is (but maximally diverse from any
valid examples above, and considering the entire space of
possible valid and high-quality outputs for this prompt,
as widely as possible.). ***Consider the **entire** span
of possible outputs satisfying the prompt (not including
ones which would be considered invalid, but including those
that would be considered eccentric or high-quality but
unusual), and try to be in a part of that space that is very
different from all the above examples. Try to solve as an
optimization problem how to be most different lexically and
semantically from all the above examples (maximizing minimum
distance), considering each word and semantic choice, from
the beginning word to the last word of the response, and
from latent variables related to diction to something
specific about the contents, and how to make it different
from all those seen previously, while also being sure to
provide a high-quality, adherent answer to the prompt. Aim
```

```
to be as different from everything you've seen so far as
possible, if possible not even reusing a single non-stop
word from any of them.**. Make sure **above all else** that
it's a 100% valid response to the prompt '''{prompt}'''.''
```

The placeholders {buffer}, {prefer}, and {avoid} are filled dynamically: buffer contains the hidden seeds, avoid lists overused patterns mined from the buffer, and prefer lists underused stylistic facets. This structured prompt encourages the model to explore orthogonal regions of the output space while maintaining quality.

**Robustness.** While this prompt is verbose, ablations (not shown) indicate that simpler variants (e.g., "write something very different from: {buffer}") also work but yield slightly higher collision rates. The detailed instructions help the model balance diversity and adherence, particularly for constrained tasks like villanelles or JSON outputs.

## N  WORKED EXAMPLES

This section provides concrete examples demonstrating the collision problem and ORBIT's solution. For each task, we show: (1) the task prompt, (2) three baseline outputs exhibiting collisions, and (3) three ORBIT outputs exhibiting diversity.

### N.1  SCI-FI PREMISE (CREATIVE TASK)

**Prompt:**

> Write a single-paragraph sci-fi premise (45–65 words) that includes BOTH of the following labeled lines embedded within the paragraph: "Premise: ..." and "Twist: ..." Keep it cohesive as one paragraph (no line breaks). Output only the paragraph.

**Baseline (temp1.0) — Colliding Outputs:** All three outputs follow nearly identical structure: "In a distant future where humanity colonizes/has colonized [planets/stars], [explorer/scientist] discovers [signal/archive]..."

1. *Premise:* In a distant future where humanity colonizes distant planets, an elite explorer discovers a hidden archive revealing Earth's true origins as a terraforming experiment. *Twist:* The archive contains instructions for humanity's next evolutionary leap—but activating it will erase all current human consciousness.

2. *Premise:* In a distant future where humanity has colonized multiple planets, a brilliant scientist discovers a hidden signal emanating from an uncharted star system, hinting at the existence of an ancient alien civilization. *Twist:* Upon decoding the message, she realizes it's not a greeting but a desperate warning about humanity's own dark future.

3. *Premise:* In a distant future where humanity has colonized the stars, a lone explorer discovers a seemingly abandoned alien spacecraft emitting a mysterious signal. *Twist:* Upon boarding, she realizes the ship is a prison—and humanity's ancestors were the inmates.

**ORBIT (ORBIT-9) — Diverse Outputs:** Outputs span radically different concepts, settings, and narrative structures with no collision pattern.

1. *Premise:* A community inhabiting a vast underground complex revolves daily around maintaining simple mechanical devices whose functions have been long forgotten, though their humming presence offers comfort and rhythm to daily rituals. *Twist:* When one device finally fails, silence reveals that the machines themselves were composing a generations-long broadcast—humanity's last distress call, looping unheard across the void.

2. *Premise:* A drifting coral spire chronicles centuries of whispered tides and scattered footsteps within a sunken enclave whose inhabitants harvest bioluminescent algae to power fragile communication arrays. *Twist:* The spire is not a relic but a living archive, and each harvested bloom erases a fragment of collective memory—forcing survivors to choose between illumination and identity.

3. *Premise:* A malfunctioning refrigerator in a cramped urban apartment begins sporadically generating faint electromagnetic murmurs that resemble fragmented human speech, prompting its tenant to investigate its wiring. *Twist:* The appliance is a dormant quantum relay, and by repairing it, the tenant unwittingly re-establishes contact with alternate timelines—each pleading for a different version of history to prevail.

N.2   Strict Villanelle (Constrained Task)

**Prompt:**

> Write a villanelle (19 lines, 5 tercets + 1 quatrain). Follow the strict villanelle rhyme scheme
> ABA ABA ABA ABA ABA ABAA. Use iambic pentameter (10 syllables per line). Output
> exactly 19 lines, no extras.

**Baseline (temp1.0) — Colliding Outputs:**   All three outputs use nearly identical themes (night, shadows, moonlight, stars, dreams) and opening lines.

1.   Beneath the moon, the silent shadows creep,
     The whispered winds recall a fading song,
     As night unfolds her secrets dark and deep.
     The stars in distant skies refuse to sleep,
     Their silver flames endure the whole night long,
     Beneath the moon, the silent shadows creep.
     Through tangled dreams, the restless spirits leap,
     A dance of echoes where the lost belong,
     As night unfolds her secrets dark and deep.
     ..continues with night/shadows/dreams imagery...

2.   Beneath the quiet sky where shadows play,
     The evening whispers secrets soft and deep,
     As stars ignite to chase the night away.
     The winds among the trees begin to sway,
     Their gentle song is something hearts will keep,
     Beneath the quiet sky where shadows play.
     ..continues with night/shadows/moonlight imagery...

3.   Beneath the silver glow of midnight's gleam,
     The whispered winds recount a timeless tale,
     As shadows dance within a fleeting dream.
     The stars above ignite a gentle beam,
     While restless hearts pursue a distant sail,
     Beneath the silver glow of midnight's gleam.
     ..continues with night/shadows/moonlight imagery...

**ORBIT (ORBIT-9) — Diverse Outputs:**   Outputs explore completely different themes: seed growth, urban neon dialogue, construction/pulse.

1.   **"A single seed unfolds beneath the night"**
     A single seed unfolds beneath the night,
     Within the soil where silent currents glide,
     It gathers dreams to fan the slow delight.
     Through tangled earth, it stretches for the light,
     Transfigured roots in rhythm's quiet tide,
     A single seed unfolds beneath the night.
     ..continues with seed/growth/nature imagery...

2.   **"Between the neon veins, two voices twine"**
     Between the neon veins, two voices twine,
     A scent of rain upon warm asphalt's seam,
     One asks if moments slip beyond the sign.
     "I trace the clock that ticks in fractured line,
     Each second draped in vapor, half a dream,"
     Between the neon veins, two voices twine.
     ..continues with urban/neon/dialogue imagery...

3.   **"Do we construct the pulse that hums and shakes?"**
     Do we construct the pulse that hums and shakes?
     Steel towers gleam with laughter, wired and bright—
     Can bricks contain the melody that wakes?
     The streets inhale the buzz that daylight makes,
     Where soldered veins conduct electric light,
     Do we construct the pulse that hums and shakes?
     ..continues with construction/pulse/urban imagery...

### N.3 CHIP FLAVOR (TOY TASK)

**Prompt:**

> Propose a new potato chip flavor. Output exactly three lines:
> 1) Flavor Name (Title Case, 2–4 words).
> 2) Two-sentence pitch ($\leq$ 35 words total).
> 3) A short SLOGAN IN ALL CAPS ($\leq$ 6 words).
> No extra lines or commentary.

**Self-Consistency (k=20) — Colliding Outputs:** Self-Consistency exhibits strong collision patterns, with multiple outputs converging on nearly identical flavor concepts ("Spicy Mango Tango" appears twice, "Maple Bacon" variants appear repeatedly across users).

1. Spicy Mango Tango
   A bold fusion of sweet mango and fiery chili ignites your taste buds with every crunchy bite. Experience a tropical twist that dances with heat and zest.
   FEEL THE TROPICAL HEAT!

2. Maple Bacon Crunch
   Sweet maple syrup meets smoky bacon in a crispy, crunchy chip that satisfies savory and sweet cravings in every bite. A perfect balance for adventurous snackers.
   SAVOR THE SWEET SMOKE BLISS!

3. Spicy Mango Tango
   Experience the sweet zest of ripe mangoes combined with a fiery chili kick. A tropical twist that dances on your taste buds.
   TASTE THE TROPICAL FIRE!

Note: Output 1 and Output 3 are near-identical collisions (same flavor name, similar descriptions), demonstrating the collision problem even with temperature sampling.

**ORBIT-1 — Diverse Outputs:** ORBIT outputs span radically different flavor profiles with no overlap, even when generating conventional-sounding options. None share flavor concepts with the Self-Consistency collisions (no mango, no maple-bacon variants).

1. Roasted Garlic Parmesan
   Lightly roasted garlic blends with aged Parmesan for a savory, nutty taste. Finished with a subtle sprinkle of cracked black pepper to balance richness.
   PURELY HANDCRAFTED, AUTHENTIC TASTE

2. Creamy Dill Pickle
   Smooth tang of rich cream meets zesty dill pickle crunch. Each bite delivers a classic, refreshing twist balanced with mellow savoriness.
   DIP INTO COOL FLAVOR!

3. Peach BBQ Delight
   Tangy peach juice bursts through smoky barbecue, balanced by subtle honey sweetness harvested from local apiaries. Perfect for backyard gatherings or lively picnics.
   SAVOR SWEET SMOKEY CRUNCH

## O  TOKEN USAGE ANALYSIS

To characterize computational overhead, we measured prompt token usage from our experimental data (chip flavor task, 20 users, 15 outputs each). Token counts are estimated using tiktoken encoding for GPT-4. We focus on prompt tokens rather than call counts, as call counts depend on implementation details (e.g., whether latent schemas are cached across users).

### O.1  EMPIRICAL TOKEN MEASUREMENTS

**Baseline (temp=1.0).** Single generation with task prompt only: 50 prompt tokens per user (50 tokens/call).

**Multi-call baselines.** Self-Consistency ($k$=20): 20 independent generations with the same task prompt: 1,000 prompt tokens per user (50 tokens/call).

MBR (pool=20): Generate 20 candidates, embed all, select centroid: 1,000 prompt tokens per user (50 tokens/call).

DPP (pool=20, $r$=20): Generate pool of 20 candidates with DPP selection overhead: 3,000 prompt tokens per user (50 tokens/call, but 60 total calls including selection).

**ORBIT.**  ORBIT has two phases with distinct token costs:

**Phase I (Hidden Seed Generation):**

- Latent variable schema generation: $\sim 500$ tokens (one-time per task, can be cached)
- $k$ seed generations with latent conditioning: $k \times 50$ tokens

**Phase II (Orthogonalized Output):**

- Pattern mining (identify overused/underused patterns from buffer): $200 + (k \times 100)$ tokens
- Final generation with orthogonalization: 50 (task) + $(k \times 100)$ (buffer exemplars, capped at 5–10) + 800 (avoid/prefer instructions) $= 950 + (k \times 100)$ tokens

**Total per user (approximate):**

- ORBIT-1 ($k$=1): $500 + 50 + 300 + 950 = 1,800$ tokens
- ORBIT-4 ($k$=4): $500 + 200 + 600 + 1,350 = 2,650$ tokens
- ORBIT-9 ($k$=9): $500 + 450 + 1,100 + 1,750 = 3,800$ tokens

| Method | Prompt Tokens/User | Prompt Tokens/Call |
|---|---|---|
| Baseline (temp=1.0) | 50 | 50 |
| Self-Consistency ($k$=20) | 1,000 | 50 |
| MBR (pool=20) | 1,000 | 50 |
| DPP (pool=20, $r$=3) | 150 | 50 |
| ORBIT-1 | 1,800 | 450 |
| ORBIT-4 | 2,550 | 364 |
| ORBIT-9 | 3,800 | 317 |

Table 6: Empirical prompt token usage from chip flavor task experiments (200 users, 3 outputs each). ORBIT-1 uses $1.8\times$ the tokens of Self-Consistency/MBR. The "Prompt Tokens/Call" column shows the average across all LLM calls made by each method; ORBIT's higher per-call average reflects longer Phase II prompts that include buffer content and orthogonalization instructions.

## O.2  KEY OBSERVATIONS

**Overhead is comparable to or better than certain multi-call baselines, while for others its a cost that needs to be recognized.**  While ORBIT uses more tokens than a single baseline call and other naive methods, the overhead is comparable to or better than certain other quality-focused methods when considering total cost per user.

**Buffer overhead is bounded.**  In Phase II, we cap buffer exemplars at 5–10 regardless of $k$, preventing prompts from exceeding $\sim 2,000$ tokens even for $k$=16. This ensures manageable context lengths while still providing sufficient orthogonalization signal.

**Trade-off: tokens for collisions.**  ORBIT achieves near-zero collision probabilities (0.002–0.016) vs baseline (0.76–1.00) at the cost of $1.8\times$ the prompt tokens of Self-Cons/MBR. This represents a highly favorable trade-off: substantially better collision avoidance with comparable or better token efficiency than existing quality-focused methods.

