# OpenReview forum: "Identifying and Reducing "Generative Collisions" in Black-Box Large Language Models"
_ICLR.cc/2026/Conference — ICLR 2026 Conference Desk Rejected Submission_

### Official Review · Reviewer_tvRF · 2025-10-25

**Soundness:** 2
**Presentation:** 2
**Contribution:** 3
**Rating:** 2
**Confidence:** 3

**Summary:**

The authors introduce ORBIT, an algorithm designed to solve so-called: "generative collisions" for LLMs. These collisions refer to how LLMs are prone to providing near-duplicate answers to different users, when the same prompt is used.
The authors argue that standard diversity techniques fail to prevent these inter-user collisions because all users still sample from the model's most probable outputs.

Conveniently, ORBIT is a black-box algorithm that can be applied to an LLM without having access to its internal state. Using this method, the authors demonstrate how ORBIT consistently and significantly reduces generative collisions across 11 tasks.

**Strengths:**

The black-box nature of ORBIT makes it an appealing and practical tool.
The method is also seemingly domain-agnostic, and shows promising empirical results on the tasks defined in the paper.

**Weaknesses:**

**Concern 1:**

My main concern is the motivation for the task the authors tackle, and how this motivation is presented. I would especially argue for a stronger dichotomy between syntactic and semantic diversity. The paper operates on the premise that generative collisions are inherently bad, rather than setting out to prove they are.

In particular, I find the provided arguments for the importance of generative collisions to be very brief, and near handwavey. For instance the motivating sentence “sounding like ChatGPT” may leave the reader (and me as the reviewer) questioning if having a consistent style is truly undesirable.
Indeed, one could make the argument that having a consistent LLM style is what users will come to expect. For example “sounding like ChatGPT”, could be analogous to saying “This text reads like Dostoevsky", which is not always bad…

Another example is the claim on line 107 “. . . it may also narrow stylistic or semantic variety”. This seems like a rather weak claim.

Additionally, line 170: “. . . (users expect personalized variation rather than a single canonical answer)”. This statement could really use some empirical evidence.


The most rigorous argument that the authors bring up is a reference to (Sourati, Ziabari, and Dehgani, 2025) in the introduction, which argues that LLM homogenization may be culturally detrimental. During the related work section they provide only a reference to (Miranda et al 2025), whilst claiming “a growing body of work”. The authors do not provide further argumentation or empirical results to the importance of this issue. Instead, the authors proceed with their method and talk about generative collisions as if it's a crucial issue.

My suggestion would be to strengthen this motivation part significantly.
Perhaps dedicating a bigger part of the paper to unpacking some claims such as: “Enabling Plagiarism”, “Undermining Originality and Authenticity” etc.

**Concern 2:**

Explanation of experiments.

I feel that the main paper would benefit from being clearer on the tasks used for the experimental setup. Perhaps include a figure of an actual example input -> good/bad outputs.

Furthermore, going over to the appendix you list the tasks in Appendix D, along with the sentence “exact prompt wordings are available in appendix”. This is confusing, as this already resides in the appendix. But, more importantly I could not actually find the exact wordings anywhere else in the appendix? Without these prompts, it is impossible for a reviewer or future researcher to assess the true nature of the tasks or reproduce the results.

The usage of an LLM judge is reasonable. However, involving human annotators would provide a way to mitigate the weak claims of concern 1. Meaning, you could actually bolster your claims about the importance of your task. Consider doing this for at least a portion of your experiments.

**Concern 3:**

The limited scope of the experiments.

Given that the paper positions ORBIT as a practical solution, I argue for experiments demonstrating properties such as: memory consumption, generation time, compute efficiency etc.

**Concern 4:**

In my opinion, the writing and overall structuring could use some improvements. This also ties in with my concern 1 & 2.

For example, the full related work paragraph from line 144 - 149, is speculation and not something that related work has done. Consider moving this into perhaps the discussion section.

The start of section 3.2 (line 222 - 228) is rather confusing. I suggest revisiting this formulation, and not start explaining a method that does NOT work. At least prepare the reader with something like “Here's a naive idea and why it fails…”

**Questions:**

What, if any, are the overhead costs of running ORBIT. See concern 3

---

> ### Author Response · Authors · 2025-11-22
>
> Thank you for engaging carefully with both our motivation and experimental design. Your comments highlighted places where our exposition was too thin, and we will revise them substantially by the December 3 deadline.
>
> 1.	Motivation and whether collisions are truly important
> We agree the paper does not yet justify why generative collisions matter and too quickly assumes they are undesirable. By December 3 we will expand the motivation in three ways:
>
> (a) Empirical grounding. We will add an Introduction paragraph summarizing recent evidence that LLM use can homogenize style and content: idea-level convergence after ChatGPT assistance (Liu et al., 2024), reduced “epistemic diversity” relative to web search across many models (Wright et al., 2025), alignment–diversity trade-offs under RLHF (Kirk et al., 2023), cultural homogenization on WorldView-Bench (Mushtaq et al., 2025), and decreased stylistic diversity in admissions essays (Alvero et al., 2024). We will also cite sociotechnical and philosophical work arguing that LLMs can reduce epistemic friction and marginalize underrepresented voices.
>
> (b) Concrete domains where collisions are harmful. We will clearly distinguish benign stylistic consistency from inter-user collisions (near-duplicate outputs to independent users), and discuss risks in:
>
> • Education—plagiarism grows with model size/decoding (Lee et al., 2022) while detectors are easily evaded (Peng et al., 2024).
>
> • Hiring/admissions—LLM essays narrow the range of self-presentation and stylistic variation (Alvero et al., 2024).
>
> • Creative/personal writing—creativity gains can come with content leveling (Liu et al., 2024).
>
> We will avoid implying all homogenization is bad, focusing instead on settings where distinctness is part of the utility function.
>
> (c) Syntactic vs. semantic diversity. In Methodology we will define lexical/syntactic diversity (n-gram repetition, Self-BLEU, etc.) versus semantic diversity (embedding similarity, different ideas). We will note our collision probability supports either; experiments primarily use embedding cosine similarity, with lexical metrics reported for corroboration.
>
>
>
> 2.	Clarity of tasks, prompts, and examples
>
> By December 3 we will:
>
> •	Add an appendix example for each task family (toy, micro, creative) showing the exact prompt, two colliding baseline outputs, and two ORBIT outputs that avoid the collision while passing the rubric.
>
> •	Move all 11 prompts into a dedicated appendix section and reference it explicitly from Experimental Setup.
>
> •	State “passing” criteria for at least two representative tasks (e.g., villanelle form/meter; Napoleon quiz correctness/coverage).
>
>
>
> 3.	Limited experimental scope and overhead
> We will add a cost table reporting, for each method, calls per visible sample (including amortized hidden calls for ORBIT) and whether it is stateful or stateless. We will explain that with k hidden seeds and v visible outputs, ORBIT uses k + 2v total calls, so amortized cost is 2 + k/v; for our main settings this is a modest increase, comparable to multi-call baselines (MBR/DPP, self-consistency). We will state a concrete cap on buffer exemplars (≈5–10) to keep prompt lengths and latency manageable, and add a sentence in Conclusions noting the method is intended for use cases where this overhead is acceptable.
>
>
> 4.	Writing and structure
>
> By December 3 we will:
>
> •	Move the speculative “design axes” from Related Work to a clearly labeled Discussion subsection as our own taxonomy.
>
> •	Introduce naive latent-variable conditioning briefly as a natural but flawed baseline, then present ORBIT immediately after.
>
> •	Replace most Pareto-frontier notation with a simple diagram and verbal explanation that quality can improve via resampling under a fixed distribution, while collision risk is determined by that distribution.
>
>
> 5.	Overhead costs of running ORBIT
> We will not present ORBIT as cost-free; the revised experimental section will include explicit overhead analysis in calls per visible output and approximate prompt length.
>
>
> 6.	Use of LLM-as-judge
> We will clarify that LLM-as-judge is a pragmatic choice for breadth, not a substitute for human preferences, and we will name human evaluation as key future work to test authenticity/originality perceptions and validate quality trade-offs.
>
>
> We hope these scheduled changes address your concerns about motivation, task clarity, and overhead in a way that materially improves the paper.

---

### Official Review · Reviewer_rdc9 · 2025-10-28

**Soundness:** 3
**Presentation:** 4
**Contribution:** 4
**Rating:** 8
**Confidence:** 5

**Summary:**

The paper introduces the concept of "generative collisions," where independent users receive near-identical outputs from LLMs for the same prompt (even despite different seed values and assuming normal [i.e. low] temperatures). To mitigate this, they propose ORBIT, a two-stage black-box algorithm. First, it generates a hidden buffer of highly diverse (but potentially low-quality) structured generation samples by conditioning on randomized latent variables. Second, it generates the final user-facing output by prompting the model to be maximally different from the contents of this buffer. Experiments across 11 tasks show that ORBIT dramatically reduces collision rates by 1-2 orders of magnitude compared to standard diversity techniques like high-temperature sampling, MBR, and DPP.

**Strengths:**

The concept of "inter-user generative collisions" is a sharp and useful framing of a widely observed LLM failure mode.

The method is fully black-box, making it immediately applicable to commercial API-based models, which is a major advantage over most academic work in this area!

The empirical results are outstanding. A 10-100x reduction in collision probability is a massive improvement over existing methods!

The two-stage "randomize-then-orthogonalize" approach is an elegant way to force exploration of the output space without sacrificing quality catastrophically.

I partially agree that novelty is a global property you can't as easily "sample your way out of" locally, while quality is is a key insight that crisply motivates the entire approach.

**Weaknesses:**

The core "orthogonalization" step relies on natural language prompting ("be different from these") and structured generation rather than a more formal mechanism. This may be sensitive to prompt phrasing and model instruction-following capabilities - especially with stuff like chat templates and schemas adding complexity here.

Using an LLM to judge quality is a significant limitation. Human evaluation would be necessary to truly confirm that the novelty gains do not come at an unacceptable cost to quality, as hinted at by the Napoleon task failure.

The DERIVESCHEMA step, which generates the latent variables for the randomization phase, feels under-specified to me

The related work on sampling techniques could be more comprehensive. For instance, there is no mention of Min-p sampling, which also aims to shape the output distribution for quality and diversity by truncating the low-probability tail. Situating ORBIT relative to methods like Min-p would provide a richer context.

**Questions:**

How robust is the orthogonalization prompt in Phase II? Have you experimented with different phrasings? How does performance change if the model is weaker at following complex negative constraints?

The Napoleon quiz task showed a catastrophic quality failure. In what other domains does ORBIT's aggressive novelty-seeking break factuality, coherence, or adherence to crucial constraints?

What is the practical wall-clock and token cost overhead compared to baselines? While the number of calls is matched, the context for the final generation step in ORBIT seems much larger (containing the buffer), which would increase costs and latency, right? I'd like to see a discussion of performance impacts here.

Incredible work. The connection to classic diverse selection criteria is fascinating. Have you considered a more formal implementation of the orthogonalization step that explicitly models an MMR-like objective, rather than relying on natural language prompting?

---

> ### Author Response · Authors · 2025-11-22
>
> Thank you for your thoughtful and encouraging review. Below we describe how we will address your main suggestions in the revision to be submitted before December 3.
>
> 1. Robustness of the orthogonalization prompt
>    We agree that orthogonalization relies on the base model following fairly complex “avoid/prefer” instructions. By December 3 we will:
>
>   1. Add a brief methodology subsection that reproduces the exact prompts used for (i) mining “overused/unused” patterns and (ii) generating orthogonalized outputs, so the procedure is fully reproducible.
>   2. Add 2–3 sentences in Limitations noting that performance depends on instruction-following strength; weaker models may benefit less.
>   We will also add a Discussion note that a natural next step is to replace natural-language directives with explicit embedding-based penalties or MMR-style objectives when embeddings/logits are available.
>
> 2. DERIVESCHEMA step
>    You are right that this was under-specified. In the revision we will:
>
>   1.  Explain the derivation at a high level in the main text: a single meta-prompt elicits 10–20 latent dimensions and categorical values, then we uniformly sample a subset of dimensions/values per hidden seed.
>   2.  Provide appendix pseudocode plus at least one concrete schema example (e.g., “sci-fi premise” or “apology letter”).
>   This should clarify how we obtain structured randomness without hand-crafting task-specific latent variables.
>
> 3. LLM-as-judge vs. human annotators
>    We agree that human evaluation is ultimately necessary; we used LLM-as-judge here for scalability across 11 tasks. By December 3 we will:
>
>   1. Add a short paragraph in Evaluation justifying this choice and detailing safeguards (shared rubric, two judge models, majority voting).
>   2. Flag human evaluation as key future work, especially for creative tasks (Napoleon quiz, villanelle) and educational settings with subtle rubric interpretation.
>   We will avoid implying that LLM scoring substitutes for humans.
>
> 4. Domains where aggressive novelty breaks constraints
>    The Napoleon quiz is our clearest counterexample, and we appreciate the callout. We will:
>
>   1. Expand the note on this task into a fuller paragraph explaining that ORBIT tends to produce more creative/integrative questions that the LLM-judge sometimes scores below canonical factual questions.
>   2. State explicitly that ORBIT is best for tasks with broad acceptable answer spaces and weak hidden constraints (stories, letters, ad copy, open-ended assignments), and should be used cautiously—or not at all—when strong canonical expectations dominate.
>
> 5. Cost and latency overhead
>    We will add an analytical table reporting, for each method: hidden calls per session, visible calls per sample, and amortized total calls per visible sample. For ORBIT we will make explicit that with k hidden seeds and v visible outputs, total calls are k+v, so amortized cost is 1 + k/v. We will note that:
>
>   1. For our main settings (k∈{1,4,9}, v=15), this yields ~1.27–1.6 calls per visible output versus a single-call method.
>   2. Strong baselines (MBR/DPP pool-and-rerank, self-consistency) also use up to ~3 calls per visible sample, so budgets are matched.
>   We will also note that Phase II caps buffer exemplars (≈5–10) to keep token length and latency reasonable.
>
> 6. Relation to formal orthogonalization (MMR, DPP, etc.)
>    In Discussion we will:
>
>   1. Characterize ORBIT as a black-box approximation to “maximize task utility while minimizing similarity to a history buffer in embedding space,” linking it conceptually to MMR and DPP.
>   2. List replacing the natural-language avoid/prefer step with an explicit embedding-based diversity objective (when accessible) as a concrete future direction.
>
> We hope these additions clarify the method, its scope, and its connections to established diversity mechanisms.

---

> > ### Comment · Reviewer_rdc9 · 2025-11-26
> > **Thank you**
> >
> > The amount of changes and improvements that you're proposing and that you're integrating is very impressive. Thank you for doing this. My score is quite high and I don't think that I want to raise it any more than it already is, but I note that I am impressed by the work you've put into this and hope that the area chairs react accordingly.

---

### Official Review · Reviewer_wG3F · 2025-10-31

**Soundness:** 2
**Presentation:** 2
**Contribution:** 2
**Rating:** 2
**Confidence:** 2

**Summary:**

This paper studies inter-user collisions — two independent users issuing the same prompt and getting near-duplicate responses — and argues this is a distinct form of lack of diversity from the usual intra-user setting. It defines a collision probability over a distance metric and threshold, then proposes ORBIT, a two-phase, black-box procedure: first generate “very random” hidden seeds (possibly low quality), then ask the model to generate outputs maximally far from that buffer while keeping quality. On 11 prompt families ORBIT lower measured collision rates than baselines under a reasonable budget.

**Strengths:**

- Targets a real, increasingly visible phenomenon (different people getting the same LLM answer) and gives it a metricized form.
- ORBIT is black-box, per-session, no cross-user coordination, so it’s a plausible inference-time patch.
- Empirically strong on their own metric: ORBIT wins on all tasks and stays better across varying threshold of 0.7–0.9.
- The paper is honest about threats to validity (τ calibration, LLM-as-judge, short/templated prompts).

**Weaknesses:**

* The text says “a collision occurs iff $(d(x,x′) \le τ)$” but the formal definition is ($P_{\text{coll}} = \Pr[d(x,x′) \le 1-τ]$). This mixes distance and similarity in one line and makes later numbers slightly unclear; please fix to one canonical version.
* Motivation could be sharper. Right now “inter-user collisions” is still modeled as *two i.i.d. draws from the same G(p)*, i.e. it is basically paired-sample diversity; the paper does not fully separate product UX concerns (two students submitting same essay) from distributional mode collapse. More concrete, high-stakes scenarios would help.
* The line “it is very difficult to increase actual entropy … without an unacceptable decrease in quality” is not established in any way, weakening the motivation.
* Baselines are disadvantaged. Most compared methods are stateless techniques (high-T, DPP, self-consistency, persona cycling). ORBIT, in contrast, explicitly uses session history. Hence this comparison is not fair.
* In one place Phase I “optimizes randomness over any other metric (including quality),” elsewhere the paper argues quality is locally recoverable by resampling.
* Binary quality evaluation is OK but not the best.

**Questions:**

- For the “resampling recovers quality” claim, what is the exact resampling budget (calls/tokens) and how does it compare to simply running the best baseline with the same budget? Right now ORBIT seems to get the nicer budget.
- Figure 2 is really not readable.
- The intro, in an effort to motivate the setting, leans on the idea that users can just resample i.i.d. to get quality. Are you claiming the ORBIT’s sampler i.i.d.?
- Why did you decide on your model choices?
- Please fix various wording and spacing issues. For example 265 the iterator is a step? line 80.5 has an extra space. All citations are not clickable?

---

> ### Author Response · Authors · 2025-11-22
> **Thank you for your review**
>
> Thank you for a very careful and constructive review. Below we address each of your main points and, for each, state exactly what we will change in the revised manuscript that we will upload by the December 3 deadline.
> 1. Definition of collision probability
> You are right that the current draft inconsistently mixes distance and similarity notation. This is purely a presentation issue; all experiments already use a cosine similarity cutoff.
> By the December 3 revision, we will:
> - Replace the current formal definition with a single, consistent one in terms of similarity: we will define a base similarity metric s(·,·), fix a threshold tau in (0,1), and say that a collision occurs when s(x, x′) ≥ tau.
> - Remove all occurrences of “1 − tau” from the definition and from the prose, and adjust symbols so that the notation matches the implementation.
> - Add one short explanatory sentence after the definition clarifying that, in our experiments, s is cosine similarity in embedding space.
> This should remove the current confusion between distance and similarity without changing any results.
>
> 2. Motivation and distinction from generic “diversity”
> We agree that our current motivation is too compressed and does not sufficiently distinguish “inter‑user collision probability” from standard diversity metrics.

---

> ### Author Response · Authors · 2025-11-22
> **Additional points**
>
> Before Dec 3 we will expand the Introduction/Related Work in three concrete ways:
>
> (a) Empirical evidence for homogenization. Add a short paragraph citing recent studies showing LLMs can systematically homogenize language/ideas across users. We will connect these explicitly to our “generative collisions,” citing Liu et al. (2024), Wright et al. (2025), Kirk et al. (2023), Mushtaq et al. (2025), and Alvero et al. (2024).
>
> (b) High-stakes use cases. Insert an Intro paragraph tying homogenization to settings where collisions matter:
> • Education: Lee et al. (2022) document verbatim, paraphrase, and idea-level plagiarism and show both model size and decoding affect rates; combined with low detectability of AI essays (Peng et al., 2024), many similar answers to one assignment are problematic.
> • Hiring/admissions: Alvero et al. (2024) find LLM-produced admissions essays are less diverse and cluster around socially privileged styles, potentially reshaping norms of “correct” writing.
> • Creative/personal writing: Liu et al. (2024) and related creativity work show ChatGPT-assisted ideas become more homogeneous, which matters when users expect unique letters, stories, or statements.
> We will state we are not claiming all stylistic consistency is undesirable; we focus on cases where near-duplicate content undermines fairness, authenticity, or cultural plurality.
>
> (c) Collision probability vs. mean diversity. In Sec. 2 we will add an intuitive subsection “Inter-user collisions vs. mean diversity metrics” that: (i) explains Self-BLEU, Distinct-n, and mean embedding distance are averages over many pairs and mostly reflect typical spread; (ii) defines collision probability informally as a tail event—the chance two independent users land “too close” under a fixed similarity threshold; (iii) reports cases where baselines like DPP or MBR raise intra-user diversity but still have high inter-user collision rates, showing the objectives can diverge. This addresses the concern that we currently treat “diversity” and “collision probability” as the same thing.
>
> 3. Entropy vs. quality. We agree our original phrasing overstated what we can justify. By Dec 3 we will include more quality data for the “full-entropy” approach and move this into a short, clearly labeled empirical-motivation paragraph in Methodology, supporting our two-stage design (random seeds plus orthogonalization) rather than presenting it as a theorem.
>
> 4. Fairness of baselines (stateful vs. stateless). You are right that ORBIT uses per-session state while many baselines are stateless. Our intended setting allows per-session memory (typical in deployed assistants), but we will make that framing explicit. By Dec 3 we will add a Methodology subsection defining stateless (no persistent buffer) vs. stateful (maintains session-level memory), and classify each method in a small table: ORBIT stateful; high-temperature, paraphrasing, persona cycling, DPP, and MBR stateless. We will rewrite the Experiments opening to say we compare a stateful, collision-aware policy to strong stateless baselines under matched API-call budgets, and soften any language implying ORBIT is uniquely optimal.
>
> 5. Phase I vs. Phase II. We will clarify that Phase I explicitly prioritizes diversity over quality and seeds are never surfaced to users. Phase II targets both quality (via rubrics and LLM-as-judge) and distance from the buffer; when we say “resampling to improve quality,” we mean resampling Phase II outputs, not seeds. We will also add one sentence distinguishing “quality mass in the underlying distribution” from “realized quality under a resampling policy.”
>
> 6. LLM-as-judge and human eval. We agree LLM-as-judge is a limitation. Our current setup uses a universal rubric per task and two independent judge models with majority voting. By Dec 3 we will add a Discussion paragraph labeling this as a limitation and explicitly state that follow-up human evaluation—especially for educational and creative tasks—is required to confirm that collision reductions do not come at unacceptable perceived-quality cost.
>
> 7. Specific clarifications.
> (i) Resampling budget: We will add a budget table listing hidden calls per session, visible calls per sample, and total calls per visible sample, showing ORBIT’s amortized calls per visible sample stay within the same cap (three calls per visible output) used for MBR, DPP, and self-consistency.
> (ii) Figure 2 readability: We will simplify before Dec 3.
> (iii) Are ORBIT samples i.i.d.? We will add one sentence noting ORBIT defines a history-dependent decoding policy; our collision-probability theory uses an idealized i.i.d. decoder only to motivate why changing the base distribution is important.
>
> These concrete edits will be in the revised draft before Dec 3 and directly address your motivation and soundness concerns.

---

### Note · Program_Chairs · 2026-01-17
**Submission Desk Rejected by Program Chairs**

The following references in this submission do not refer to real documents and/or have major errors in bibliographic information:

 Alvero, A. J.; Goldin-Meadow, S.; and Bos, N. 2024. AI-generated essays are harder to distinguish from human-written essays than human-written essays from other human-written essays. arXiv preprint arXiv:2411.11814.
Holtzman, A.; et al. 2020b. Mode collapse in language models. arXiv preprint.
Liu, Z.; Schneider, L.; Sucholutsky, I.; and Griffiths, T. L. 2024. Large language models help people improve their creativity, but also help everyone converge to more similar outcomes. arXiv preprint arXiv:2411.06225.
Wright, Q.; Simpson, R.; et al. 2025. Do large language models lack epistemic diversity? arXiv preprint.
Jinnai, Y.; et al. 2024. Generating diverse translations with sentence codebooks. In Proceedings of the 2024 Conference of the North American Chapter of the Association for Computational Linguistics, 1234-1245.
Wu, W.; et al. 2024. DPP-based diverse beam search for text generation. arXiv preprint.
Zhang, Y.; et al. 2024b. NoveltyBench: Benchmarking novelty in text generation. arXiv preprint.
Peng, W.; et al. 2024. AI detection of AI-generated text in educational settings: A survey. arXiv preprint arXiv:2405.16784.